EMBO
Molecular Medicine

# The immunoproteasome-specific inhibitor ONX 0914 reverses susceptibility to acute viral myocarditis

Nadine Althof[1,2,†], Carl Christoph Goetzke[1,†] (iD), Meike Kespohl[1,2], Karolin Voss[1], Arnd Heuser[3], Sandra Pinkert[1], Ziya Kaya[4,5], Karin Klingel[6] & Antje Beling[1,2,*] (iD)

## Abstract

Severe heart pathology upon virus infection is closely associated with the immunological equipment of the host. Since there is no specific treatment available, current research focuses on identifying new drug targets to positively modulate predisposing immune factors. Utilizing a murine model with high susceptibility to coxsackievirus B3-induced myocarditis, this study describes ONX 0914—an immunoproteasome-specific inhibitor—as highly protective during severe heart disease. Represented by reduced heart infiltration of monocytes/macrophages and diminished organ damage, ONX 0914 treatment reversed fulminant pathology. Virus-induced immune response features like overwhelming pro-inflammatory cytokine and chemokine production as well as a progressive loss of lymphocytes all being reminiscent of a sepsis-like disease course were prevented by ONX 0914. Although the viral burden was only minimally affected in highly susceptible mice, resulting maintenance of immune homeostasis improved the cardiac output, and saved animals from severe illness as well as high mortality. Altogether, this could make ONX 0914 a potent drug for the treatment of severe virus-mediated inflammation of the heart and might rank immunoproteasome inhibitors among drugs for preventing pathogen-induced immunopathology.

**Keywords** immunology and inflammation; infectious diseases; myocarditis; proteasome; virus

**Subject Categories** Cardiovascular System; Microbiology, Virology & Host Pathogen Interaction; Pharmacology & Drug Discovery

## Introduction

Myocarditis is an inflammatory condition of the heart most commonly triggered through viral infection especially in young individuals (Epelman *et al*, 2015). But despite the fact that underlying processes and mechanisms have been elucidated from different angles by basic research for several decades, there is still no specific treatment available. Further, it is unclear which circumstances either prevent infected patients from clinical manifestation or provoke severe pathology. The disease outcome may range from subclinical disease to serious illness with acute heart failure and sudden cardiac death and is mainly determined by an interplay between virus-mediated cytotoxicity and the host's immune response (Cooper, 2009; Sagar *et al*, 2012). Besides other aspects, the genetic background and associated with it the immunological equipment of the host are thought to be predisposing factors for the development of severe heart muscle inflammation. In this context, scientific research utilized diverse inbred mouse strains to mirror cardiac pathology in human patients (Corsten *et al*, 2012). Such preclinical animal models are suited to investigate the efficacy of novel immune-modulatory drug targets (Leuschner *et al*, 2015).

The proteasome—a multicatalytic protease essential for intracellular protein degradation and viability of all mammalian cells—is currently in the focus of research seeking new targets to treat cancer and inflammatory illness (Groettrup *et al*, 2010; Basler *et al*, 2013). First- and second-generation proteasome inhibitors, such as bortezomib and carfilzomib, are used for patients with multiple myeloma (Richardson *et al*, 2003, 2005; Stewart *et al*, 2015). These inhibitors target both major forms of the proteasome—the standard proteasome of all somatic cells and the immunoproteasome mainly expressed in cells of the immune system. The development of third-generation inhibitors selective for the immunoproteasome has opened up new opportunities not only for immunoproteasome

---

1 Charité – Universitätsmedizin Berlin, corporate member of Freie Universität Berlin, Humboldt-Universität zu Berlin, and Berlin Institute of Health (BIH), Institute of Biochemistry, Berlin, Germany
2 Deutsches Zentrum für Herz-Kreislauf-Forschung (DZHK), partner side Berlin, Berlin, Germany
3 Max-Delbrueck-Center for Molecular Medicine Berlin, Berlin, Germany
4 Medizinische Klinik für Innere Medizin III: Kardiologie, Angiologie und Pneumologie, Universitätsklinikum Heidelberg, Heidelberg, Germany
5 Deutsches Zentrum für Herz-Kreislauf-Forschung (DZHK), partner side Heidelberg, Heidelberg, Germany
6 Cardiopathology, Institute for Pathology and Neuropathology, University Hospital Tuebingen, Tuebingen, Germany
*Corresponding author. Tel: +49 30 450 528 187; Fax: +49 30 450 528 921; E-mail: antje.beling@charite.de
†These authors contributed equally to this work

 

research itself, but also for future clinical use. ONX 0914, originally known as PR957, is such a potent inhibitor specific for the highly active immunoproteasome subunit LMP7 (β5i; Muchamuel *et al*, 2009; Huber *et al*, 2012). Preclinical animal models utilizing this compound demonstrated the therapeutic potential of selectively targeting the immunoproteasome mainly for inflammatory disorders (Muchamuel *et al*, 2009; Basler *et al*, 2010; Ichikawa *et al*, 2012). Furthermore, the ONX 0914 analog KZR-616 has only recently entered clinical development phase I/II trials treating patients with autoimmune-triggered inflammation. These studies are based on the highly regulated immunoproteasome function of controlling cytokine production and antigen presentation (Muchamuel *et al*, 2009; Basler *et al*, 2010, 2014; Groettrup *et al*, 2010; Kalim *et al*, 2012; Mundt *et al*, 2016a), which is in turn intensively influenced through its inhibition.

However, an anti-inflammatory potential of immunoproteasome inhibition during infectious disease is still not characterized to the same extent as it is for autoimmune disease. In addition to inducing immunological protection from invading pathogens, infection can also trigger overwhelming immune-pathological responses, leading to destructive inflammation and organ damage. The importance of a balanced immunoproteasome function in counteracting infection and thereby influencing clinical signs and potentially disease outcome has been shown for different pathogens by others (Tu *et al*, 2009; Basler *et al*, 2011; Ersching *et al*, 2016; Mundt *et al*, 2016a,b) as well as for coxsackievirus B3 (CVB3)-mediated myocarditis by our own group (Opitz *et al*, 2011; Paeschke *et al*, 2016). Further, the extent of inflammation during a suspected viral myocarditis in patients is considered to be an independent negative predictor of disease outcome (Kindermann *et al*, 2008). Since excessive inflammation can be antagonized by selective immunoproteasome inhibition (Groettrup *et al*, 2010; Basler *et al*, 2013), this study aimed to investigate the disease-modifying potential of the third-generation inhibitor ONX 0914 during virus-induced myocarditis. We addressed the questions to what extent ONX 0914 treatment influences inflammation and disease outcome, and how this treatment modulates the immune response after pathogen encounter.

# Results

## ONX-0914 treatment of C57BL/6 mice affects viral myocarditis only mildly and results in intact activation of CVB3 adaptive immune responses

Following up studies on previous work resolving proteasome biology in C57BL/6 mice with hereditary low susceptibility to virus-induced myocarditis (Jakel *et al*, 2009; Opitz *et al*, 2011; Rahnefeld *et al*, 2011; Ebstein *et al*, 2013), this mouse strain was treated with ONX 0914 and infected with a cardiotropic CVB3 variant. Since complete selective inhibition of LMP7 subunits detected 3 h after ONX 0914 injection started to decline by 24 h post-treatment, ONX 0914 was administered s.c. on a daily basis. Thereby, the as robust described phenotype in terms of cardiac disease, mortality and overall condition was only mildly affected by ONX 0914 (Fig 1A and B). The day 8 p.i. status of virus-induced heart disease, which is in general a rather mild one in C57BL/6 mice (B6), was mildly influenced by ONX 0914, whereby the extent of myocardial injury represented by

the area of inflammation on heart tissue sections was increased (Fig 1C). Further, the levels of pro-inflammatory cytokines/chemokines in heart tissue were higher if affected at all compared to vehicle-treated controls (Fig 1D). Consistent with mild increase in myocardial injury, viral load of heart tissue was higher in ONX 0914-treated mice (Fig 1E), which is most likely due to ONX 0914-induced suppression of type I interferon (IFN) responses (Fig 1F). Importantly, ONX 0914-treated B6 mice were able to produce and maintain CVB3-neutralizing antibodies levels as efficiently as their vehicle-treated controls (Fig 1G). In turn, this surrogate finding reflects an effective B and CD4[+] T-cell activation upon viral infection even under the influence of ONX 0914. In order to investigate whether the development of an adequate immune memory status is established unhindered as well, on day 28 after the initial virus infection, B6 mice were challenged a second time with CVB3. Fig 1H–J demonstrates unequivocally that both test groups showed no visible or measurable signs of re-infection. Mice were in overall good condition mirrored by body weight status as a typical parameter of an ongoing CVB3 infection (Fig 1H). Heart tissue was free of infectious virus particles on day 8 after challenge (Fig 1I). Furthermore, no signs of heart muscle damage or immune cell re-infiltration were found in vehicle—as well as ONX 0914-treated B6 mice (Fig 1J).

## High susceptibility to virus-induced myocarditis in A/J mice is inverted by ONX 0914 treatment leading to preservation of cardiac output

As a next step, we investigated the impact of an ONX 0914 treatment in a highly susceptible host. In clear contrast to B6 mice, A/J mice represent the group of human patients predisposed for the development of severe acute viral cardiomyopathy (Chow *et al*, 1991; Klingel *et al*, 1992). While 60% of CVB3-infected vehicle-treated mice had succumbed by day 7 p.i., ONX 0914 treatment saved mice from severe illness and elevated survival rate up to 100% again (Fig 2A and B). In line with this, analysis of heart tissue obtained from animals sacrificed 8-day p.i. revealed distinct differences. Histological and more clearly immunohistochemical staining of heart tissue (Fig 2C) and the following quantitative evaluation of Mac-3-positive-stained area (Fig 2D) demonstrated a profound myocarditis in vehicle-treated A/J mice and in contrast to that only moderate signs of inflammation after inhibitor treatment. Consistent with previous findings (Opitz *et al*, 2011; Rahnefeld *et al*, 2014), Mac-3-positive signals were highly abundant especially within inflammatory foci during myocarditis and ONX 0914-induced effects were most impressive for these signals (Fig 2C and D). In addition, infiltrated immune cells of hearts from vehicle- and ONX 0914-treated mice were quantitatively analyzed by flow cytometry revealing a significant reduction in infiltrating myeloid CD11b[+] immune cells in general in ONX 0914-treated mice (Fig 2E) and thus corroborating reduced inflammatory organ damage under ONX 0914 influence. Importantly, reversal of susceptibility to virus-induced myocarditis by ONX 0914 was reflected by detection of additional signs of reduced immune-mediated injury such as significantly decreased expression levels of the T-cell effector molecules granzyme A, perforin-1, and IFN-γ in heart tissue (Fig 2F). Also, the overall abundance of cell death was reduced by ONX 0914 as indicated by detection of lower viability dye signal intensity in non-immune cells by flow cytometry analysis of heart tissue (Fig 2G).

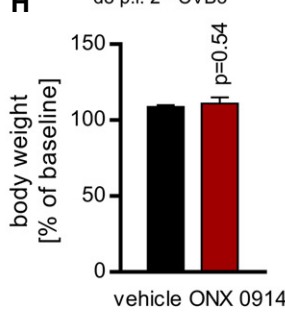
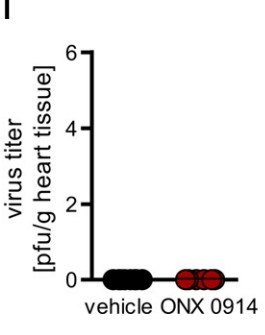
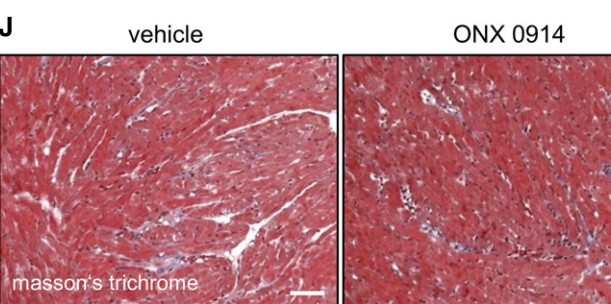

**Figure 1.**

ONX 0914 effects on cardiac performance were assessed by echocardiography during the inflammatory peak of viral myocarditis (Table 1). In vehicle-treated, infected A/J mice, the cardiac output was severely suppressed in comparison with baseline measurements [baseline: $9.9 \pm 0.6$ ml/min vs. 8-day p.i.: $5.3 \pm 0.5$ ml/min ($P < 0.05$)]. Diminished stroke volume was attributed to significant disturbances of diastolic filling as indicated by reduction in diastolic volume and increased isovolumic relaxation time, but not connected to changes in ejection fraction. Consistent with its protective immune-modulatory effects, ONX 0914 treatment impressively reversed the impaired diastolic filling found in vehicle-treated mice. Most strikingly, in comparison with vehicle treatment, cardiac output was significantly higher under ONX 0914 influence (ONX 0914: $7.7 \pm 0.6$ ml/min vs. vehicle: $5.3 \pm 0.5$ ml/min [$P < 0.05$]) and not remarkably affected by CVB3 infection at all. Furthermore, ONX 0914 beneficially affected the heart rate. In clear contrast to

**Figure 1.  Impact of ONX 0914 on CVB3 infection in C57BL/6 mice.**

A–F    Starting one day prior to CVB3 (Nancy) inoculation, C57BL/6 mice were treated daily until day 10 p.i. with either ONX 0914 or vehicle and further three times a week until day 28 p.i. Survival (A) as well as body weight (B) was monitored for 28 days. Average percentages of body weight loss relative to initial weight ± SEM are shown (vehicle *n* = 8, ONX 0914 *n* = 8). On day 8 p.i., where inflammation of the myocardium is expected to be fully developed, animals were sacrificed (C), and heart tissue was analyzed. Paraffin-embedded heart sections were stained using hematoxylin and eosin. A representative image for each group is shown. Extent of myocardial infiltration was scored microscopically, and percentage of inflamed area was assessed and calculated using ImageJ (vehicle *n* = 13, ONX 0914 *n* = 11). Total heart tissue mRNA was isolated (D), reverse transcribed and cytokine/chemokine-specific gene expression was determined by TaqMan qPCR (vehicle *n* = 10, ONX 0914 *n* = 10). Virus load (E) was determined by standard plaque assay assessing the amount of infectious virus particles (vehicle *n* = 13, ONX 0914 *n* = 10). IFN-β serum levels (F) during early infection (day 2 p.i.) were determined by ELISA (control=uninfected animals, vehicle *n* = 7, ONX 0914 *n* = 7). CVB3-neutralizing antibody titers in the serum (G) were determined after 8 and 28 days post CVB3 infection (day 8 vehicle *n* = 13, ONX 0914 *n* = 10, day 28 vehicle *n* = 8, ONX 0914 *n* = 7).

H–J    To investigate the impact of ONX 0914 treatment on developing an immunological memory status, all mice were challenged with a second CVB3 infection on day 28 after the initial one. Animals received no further ONX 0914 treatment. Proportional body weight loss on day 8 after second challenge relative to the weight on day 28 after the initial challenge (H). Viral load of heart tissue (I) was determined by plaque assay after 8 days. Additionally, paraffin-embedded heart tissue sections were prepared and Masson's trichrome staining was performed (J). Representative images for each group are shown (one experiment was carried out; vehicle *n* = 8, ONX 0914 *n* = 6).

Data information: Data are mean ± SEM. Scale bars, 60 μm. *P*-values are indicated in each graph. Survival curves were estimated from the Kaplan–Meier procedure with the log-rank (Mantel–Cox). For body weight course, two-way ANOVA was used. Otherwise, unpaired *t*-test was conducted.

vehicle treatment, ONX 0914 treatment led to intact diastolic filling (ONX 0914: 36.2 ± 1.9 μl vs. vehicle: 29.1 ± 1.5 μl [*P* < 0.01]) yielding no significant deterioration during infection.

Since direct cytolysis of cardiomyocytes by the virus itself is closely and causally connected with infiltration of immune cells during acute state of myocarditis (Althof *et al*, 2014), next we investigated whether ONX 0914-induced amelioration of myocardial injury as shown in Fig 2 and reflected by improved cardiac output (Table 1) might also involve alteration of the viral burden. Therefore, viral titers of heart tissue usually peaking around 4–6 days after initial virus inoculation (Opavsky *et al*, 2002) were determined throughout disease course. We found reduced coxsackievirus genome and titers under ONX 0914 influence shortly after virus inoculation (day 2 p.i.), but at any point in time beyond infectious particles were not affected by ONX 0914 (Fig 3A and B). Of note, ONX 0914 did not influence the virus replication cycle directly (Fig EV1B). Since the pancreas and not the heart is the primary organ in which CVB3 quickly replicates to high titers (Klingel *et al*, 1992, 1996), we questioned whether reduced viral burden 2 days after infection in heart tissue might be attributed to an altered viral load in pancreas. We found no effect of ONX 0914 treatment on pancreatic viral titers (Fig 3C). In contrast to that and consistent with observations for heart tissue, the viral burden was reduced on day 2 p.i. in spleen and serum as well (Fig 3D and E). Altogether, these findings indicate that a temporarily reduced viral burden in heart, spleen, and serum most likely reflects slightly altered viremia, which in turn could be the result of a diminished immune-mediated destruction of pancreatic tissue. And indeed, we found altered signs of pancreas destruction such as reduced serum lipase activity in ONX 0914-treated mice (Fig EV1B and C). Taken together, since ONX 0914 had overall no effect on the viral burden in heart and pancreas at the respective organ-specific peak of viral cytotoxicity, ONX 0914-induced protection from virus-induced pathology in A/J mice appears not to be attributed to a major antiviral effect and its positive sequels.

**ONX 0914 influences lymphocyte abundance during virus infection**

We and others have demonstrated that high susceptibility to viral myocarditis involves inefficient activation of type I IFN responses (Wessely *et al*, 2001; Jakel *et al*, 2009; Rahnefeld *et al*, 2011, 2014;

Epelman *et al*, 2015). In line with this, we found IFN-β production in CVB3-infected A/J mice to be severely impaired in general. ONX 0914 treatment exerted no significant additive detrimental effect on systemic IFN-β production or tissue-resident type I IFN responses (Fig EV3). In order to investigate, which immune factors might contribute to ONX 0914-induced protection from virus-induced immunopathology, we next elucidated the effect of ONX 0914 on the immune cell equipment of the host during CVB3 infection. Immunoproteasome function is known to regulate lymphocyte survival, proliferation, and differentiation as well as function (Fehling *et al*, 1994; Hensley *et al*, 2010; Moebius *et al*, 2010; Kalim *et al*, 2012) all being closely connected to antigen presentation (Kincaid *et al*, 2012). Based on this, we first investigated the immune status mice exhibit in terms of lymphocyte abundance, both right after ONX 0914 application and in addition after a longer period of inhibitor exposure that corresponded to the time after CVB3 infection at which myocarditis is fully developed. This allowed us to determine direct ONX 0914-mediated effects apart from virus infection. Figure 4A and B demonstrates significantly decreased numbers of splenic T lymphocytes after single inhibitor treatment and a reduction in splenic B-cell count after a 9-day treatment. Next, we analyzed blood as well as the spleen as a representative lymphoid organ regarding lymphocyte dissemination upon pathogen challenge. During CVB3 infection course, vehicle-treated A/J mice suffered a marked depletion of lymphocytes in the periphery, which was completely prevented under ONX 0914 influence (Fig 4C). Further, during early stages of disease, spleens of compound-treated animals were populated by significantly more T lymphocytes compared to controls (Fig 4D). Hence, the restrained lymphocyte depletion found in infected, ONX 0914-treated mice at early stages of disease is most likely not directly mediated by ONX 0914, but rather attributed to secondary ONX 0914-mediated mechanisms of the systemic immune response during infection. By day 8 p.i., T-cell count in spleen equalized to comparable numbers in both vehicle- and ONX 0914-treated mice. However, splenic B-cell numbers were slightly reduced under inhibitor influence (Fig 4E). As B-cell function is indispensable for humoral antiviral immunity during an infection with CVB3 (Mena *et al*, 1999), we next investigated CVB3-neutralizing antibodies and found increased titers under ONX 0914 influence 8-day p.i. (Fig 4F) indicating intact or even improved activation of antiviral humoral immunity in ONX 0914-treated A/J mice.

**Figure 2.  Impact of ONX 0914 on CVB3 infection in A/J mice.**

A–G  A/J mice were infected with $10^4$ PFU of CVB3 (Nancy). ONX 0914 or vehicle treatment was carried out daily, starting one day prior to virus inoculation. Proportional survival of animals during the first 8 days of infection (A) was implemented into a Kaplan–Meier survival curve (E1 + E2; vehicle $n = 12$, ONX $n = 12$; log-rank test). Body weight (B) was monitored for an 8-day period. For the indicated time points p.i., average percentage of weight loss relative to the initial value ± SEM is shown (E3; vehicle $n = 15$, ONX $n = 15$; two-way ANOVA test followed by Bonferroni multiple comparison). On day 8 p.i., animals were sacrificed and heart tissue was analyzed. Paraffin-embedded tissue sections were prepared and differently stained (C). Upper row shows representative images for classical hematoxylin and eosin staining. Middle and bottom row display images of immunohistochemical staining specific for either Mac3$^+$ monocytes/macrophages or CD3$^+$ T lymphocytes, respectively (E1 + E2; vehicle $n = 4$, ONX $n = 12$). Scale bars, 60 μm. Additionally, proportional area infiltrated by Mac3$^+$ monocytes/macrophages (D) was assessed and calculated using ImageJ (means + SEM; E1 + E2; vehicle $n = 4$, ONX $n = 12$; unpaired *t*-test). Immune cells of the heart were stained with fluorochrome-labeled antibodies (E). Number of CD11b$^+$ myeloid cells per milligram heart tissue was determined by flow cytometry (means + SEM; E3; vehicle $n = 3$, ONX $n = 8$; unpaired *t*-test). Further, total heart tissue mRNA (day 8 p.i.) was used to determine T-cell effector molecules (F) by TaqMan qPCR. Means of $2^{-\Delta Ct}$ + SEM are shown (E1 + E2; vehicle $n = 4$, ONX $n = 12$; unpaired *t*-tests). Non-immune cells of the heart (CD45$^-$) (G) were stained with Fixable Viability Dye eFluor 780 (eBioscience), and mean fluorescence intensity (MFI) was determined by flow cytometry (means + SEM; E3; vehicle $n = 3$, ONX $n = 8$; unpaired *t*-test). *P*-values are indicated in each graph.

## ONX 0914 regulates dissemination and function of neutrophils

Although antiviral immune responses are dominated by lymphocytes, the innate immune system also plays a central role in the host antiviral response helping to shape and control the ensuing adaptive response (Jenne & Kubes, 2015). Hence, we investigated whether abundance and function of innate immune cells might be influenced by ONX 0914. We found that myeloid cells in general and neutrophils in particular were released from the bone marrow quickly after ONX 0914 application. Dissemination resulted in higher numbers of neutrophils in blood and spleen after a single ONX 0914 injection (Fig 5A). ONX 0914-induced neutrophilia was also evident after a

**Table 1. Analysis of cardiac function after ONX 0914 treatment in CVB3-infected A/J mice.**

| | Vehicle | | ONX 0914 | |
|---|---|---|---|---|
| | **Baseline** | **Day 8 CVB3** | **Baseline** | **Day 8 CVB3** |
| CO [ml/min] | 9.9 ± 0.6 | 5.3 ± 0.5 | 8.8 ± 0.6 | 7.7 ± 0.6[a] |
| Vol-d [µl] | 42.4 ± 1.4 | 29.1 ± 1.5 | 38.7 ± 1.4 | 36.2 ± 1.9[a] |
| Vol-s [µl] | 19.7 ± 0.8 | 14.3 ± 1.1 | 17.2 ± 1.1 | 19.3 ± 1.2[a] |
| SV [µl] | 22.7 ± 1.2 | 14.9 ± 1.0 | 22.0 ± 1.3 | 16.9 ± 1.0 |
| LVEF | 53.2 ± 1.8 | 51.6 ± 2.5 | 54.8 ± 2.5 | 46.7 ± 1.5 |
| LV-d [mm] | 3.6 ± 0.1 | 3.3 ± 0.1 | 3.5 ± 0.1 | 3.3 ± 0.1 |
| LV-s [mm] | 2.6 ± 0.1 | 2.5 ± 0.1 | 2.5 ± 0.1 | 2.7 ± 0.1 |
| IVRT [ms] | 16.9 ± 0.6 | 20.4 ± 0.7 | 16.6 ± 0.6 | 16.4 ± 0.7[a] |
| MV$_{decel}$ [ms] | 23.5 ± 1.0 | 31.7 ± 1.5 | 27.4 ± 1.3 | 22.7 ± 1.4[a] |
| Heart rate [bpm] | 435 ± 12 | 352 ± 20 | 399 ± 8 | 446 ± 17[a] |

CO, cardiac output; Vol-d, end-diastolic volume; Vol-s, end-systolic volume; SV, stroke volume; LVEF, left ventricular ejection fraction; LV-d, left ventricle internal diameter at diastole; LV-s, left ventricle internal diameter at systole; IVRT, isovolumic relaxation time; MV$_{decel}$, deceleration time (IVRT, MV$_{decel}$ were determined by pulse-wave Doppler at mitral valve); bpm, beats per minute.
Cardiac function at baseline was assessed by echocardiography directly prior to the second ONX 0914 injection in A/J mice (baseline). The measurement was followed by CVB3 inoculation on the same day. Mice were allocated to respective groups: vehicle n = 15; ONX 0914 n = 15. On day 8 p.i., echocardiography was repeated (vehicle n = 14; ONX 0914 n = 14 mice). Data are mean values ± SEM. Repeated measurements two-way ANOVA was performed followed by a Sidak's multiple comparison test.
[a]Significant differences (*P* < 0.05) between vehicle and ONX 0914-treated mice at day 8 after infection as determined by *post hoc* analysis. There were no significant differences between vehicle and ONX 0914-treated mice at baseline.

longer period of ONX 0914 treatment (Fig 5B). Upon CVB3 infection, ONX 0914-induced neutrophilia further increased in blood (1- and 2-day p.i.). Control animals also reacted to virus infection with a neutrophilia peaking around day 2 (Fig 5C). Further, ONX 0914 treatment during myocarditis leads to elevated neutrophil counts in spleen tissue persisting at least until complete evolvement of acute myocarditis (8-day p.i.; Fig 5D). In order to get a more elaborated impression of how neutrophils become activated under the influence of ONX 0914 treatment, we next analyzed the neutrophil population regarding their expression level of activation markers. CD18 (β2 integrin), as part of the ICAM-binding complex Mac-1, was significantly upregulated, whereas CD62L expression was markedly reduced upon ONX 0914 treatment, all being in line with an activation of neutrophils (Fig 5E). Also, ONX 0914 had a significantly positive impact on phagocytosis capacity of neutrophils (Fig 5F). Since ONX 0914 treatment exerted such impressive effects on neutrophils, we questioned whether these cells might be involved in the control of CVB3. A unique neutrophil effector mechanism is the formation of neutrophil extracellular traps (NETs). NETs are structures comprised of a sticky, complex mesh of decondensed strands of nuclear DNA released into the extracellular environment with protective for some enveloped viruses (Jenne & Kubes, 2015). Hence, co-culture experiments using NETs-forming neutrophils and CVB3 were conducted. Figure 5G demonstrates that NET formation had no influence on the

resulting infection of NET-exposed CVB3 in HeLa cells as represented by similar viral genome and capsid protein abundance as well as unhindered formation of infectious CVB3 particles. To obtain a more elaborate picture on the biological relevance of neutrophils during CVB3 infection, we depleted these cells using a Ly6G-specific antibody in A/J mice. Upon CVB3 infection, neutrophil-depleted mice were in an overall good condition regarding body weight loss and survival. Most importantly, virus load, which was determined for heart, pancreas, and spleen tissue after 2, 3, and 8 days, was found to be within the same range as that of the respective control group (Figs 5H and EV4). Altogether, these data suggest that neutrophil function is most likely not essential for CVB3 control and indicate that ONX 0914-induced neutrophilia is not decisive for protection against immunopathology.

**ONX 0914 regulates dissemination of monocytes/macrophages**

Monocytes and macrophages, which are centrally involved in mediating tissue damage and were reduced upon ONX 0914 treatment in inflammatory heart disease (Fig 2), originate, like neutrophils, from hematopoietic stem cells or subsequent progenitor stages. To investigate whether reduced infiltration into heart tissue may be the result of altered mobilization of these cells, ONX 0914-induced effects on the abundance of two different subsets of monocytes expressing either high or low/medium levels of Ly6C as well as macrophage counts were determined (Fig EV2). ONX 0914 treatment increased especially the number of blood and splenic Ly6C$^{high}$ inflammatory monocytes significantly (Fig 6A). Mononuclear phagocytes as represented by macrophages might be derived from inflammatory monocytes during infection (Ginhoux & Jung, 2014). As demonstrated for neutrophils, ONX 0914 had a significantly positive impact on phagocytosis capacity of macrophages as well (Fig 6B). As a next step, we investigated ONX 0914-induced effects on monocytes/macrophages during infection and found a substantially pronounced impact of the inhibitor. During the course of CVB3 infection, ONX 0914 treatment led to elevated counts particularly of monocytes in spleen tissue (Fig 6C) resulting in an increased number of Ly6C$^{high}$ monocytes at the stage of complete evolvement of acute myocarditis (8-day p.i.; Fig 6D). Taken together, ONX 0914 mobilized monocytes from the bone marrow during viral infection.

**ONX 0914 treatment results in suppressed pro-inflammatory cytokine/chemokine production during CVB3 infection**

The demonstrated ONX 0914-mediated effects on the different immune cell subsets during infection are indeed in agreement with a functional activation of adaptive immune responses (Figs 1G–I and 4F). However, these findings did not fully explain the improved survival, attenuated heart inflammation, and maintained cardiac output that we found in ONX 0914-treated mice. Pro-inflammatory cytokine and chemokine are critical mediators that regulate migration and infiltration of immune cells. Moreover, for most of these molecules, experimental evidence illustrates a disease-modifying function during myocarditis (Corsten *et al*, 2012). Hence, next we focused on the impact of ONX 0914 treatment on the production of central inflammatory mediators like TNF-α, IL-6, IL-1β, MCP-1, IP-10, and MIP-2α during disease. Despite increased infiltration of

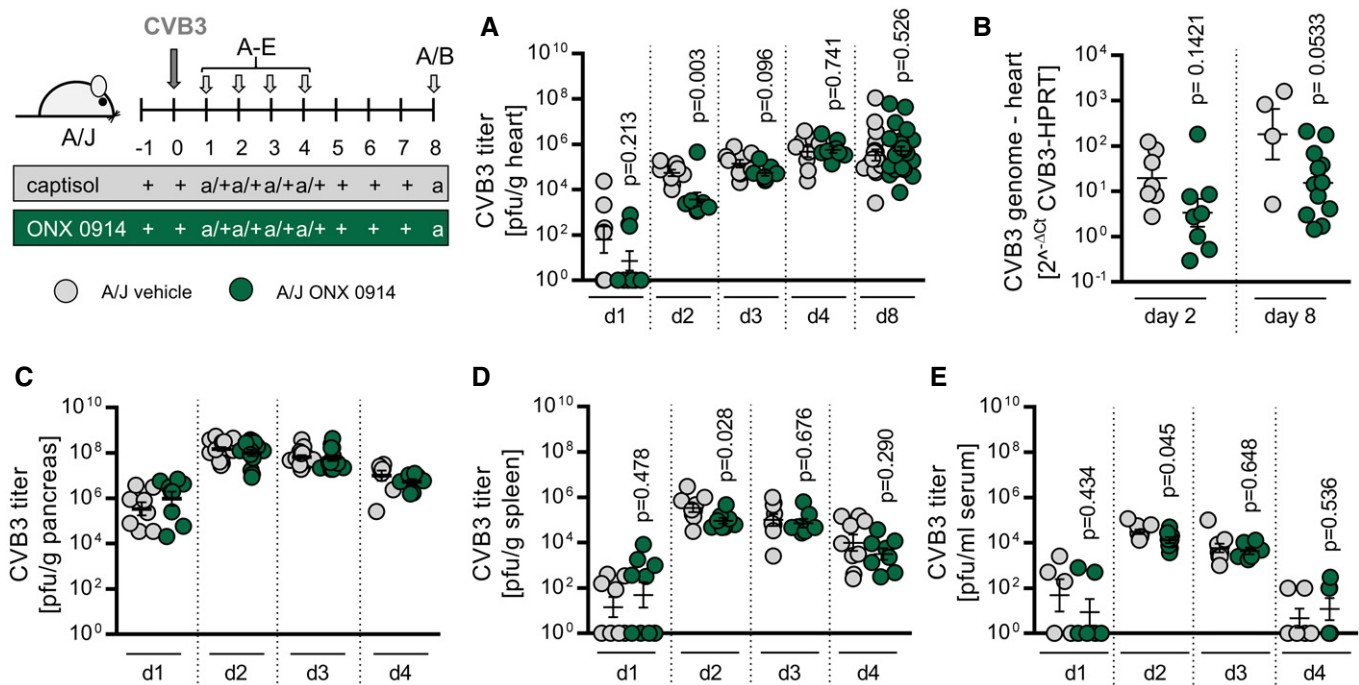

**Figure 3. Viral burden during course of CVB3 infection in ONX 0914-treated A/J mice.**

A–E  A/J mice were infected with $10^4$ PFU CVB3 (Nancy). ONX 0914 or vehicle treatment was carried out daily, starting one day prior to virus inoculation. Animals were sacrificed at the indicated time p.i.: Heart tissue, pancreas, and spleen were isolated, and serum obtained. Viral load was determined by standard plaque assay (A, C–E) assessing the amount of infectious virus particles as well as by TaqMan qPCR (B), detecting CVB3 genome copy levels in total myocardial RNA (day 8 p.i.: E1 + E2; vehicle $n$ = 4, ONX $n$ = 12). Transformed means ± SEM are presented. Unpaired $t$-tests were performed. $P$-values are indicated in each graph.

myeloid cells that majorly contribute to cytokine production into splenic tissue, the expression of respective pro-inflammatory cytokines/chemokines was suppressed under ONX 0914 influence (Fig 7A). This anti-inflammatory activity upon immunoproteasome inhibition was corroborated by detection of significantly reduced systemic amounts of TNF-α, IL-6, IL-1β, MCP-1, and MIP-2α in ONX 0914-treated mice 3 days after virus inoculation, respectively (Fig 7B). Further, also at the stage of fully evolved inflammation of heart tissue (day 8), the local expression of the respective cytokines/chemokines was substantially reduced by ONX 0914 (Fig 7C). The overwhelming cytokine response which infected vehicle-treated mice developed is in accordance with impaired diastolic filling, high-grade infiltration, reduction in the cardiac output as well as high mortality and overall reminiscent of a distributive shock condition in this group.

Since ONX 0914-induced suppression of cytokine/chemokine production was identified as a major effector of overall diminished pathology, we focused on this aspect in more detail. Thereby, experimentally bone marrow-derived macrophages (BMM) were challenged with artificial single-stranded RNA—a Toll-like receptor 7 (TLR7) agonist—mirroring natural conditions under which intracellular coxsackievirus RNA genomes serve as pathogen-associated molecular patterns (PAMPs) and induce strong cytokine expression in monocytes/macrophages provided by the bone marrow in response to virus encounter. TLR7-activated BMM responded with increased TNF-α, IL-1β, IL-6, MCP-1, IP-10, and MIP-2α production

(Fig 8). TLR7-engagement allows binding between TLR7 and MyD88, which then induces mitogen-activated protein kinase (MAPkinase) signaling (Blasius & Beutler, 2010). We investigated whether MAPkinases might be involved and initially used specific inhibitors of the three major MAPkinases EKR1/2, p38, and JNK to elucidate a putative contribution of the respective kinases to TLR7-induced cytokine/chemokine production. Fig 8A illustrates a role of MAPkinase activation particularly for IL-6 and IL-1β production. Moreover, we found that SP600125, which interferes with JNK/p38 phosphorylation, also reduced TNF-α expression and MIP-2α expression (Fig 8A). Next, we questioned whether as observed for spleen and heart tissue during infection, ONX 0914 treatment influences cytokine/chemokine production also in TLR7-activated BMM. We found a profoundly suppressed cytokine/chemokine mRNA induction in cells that had been treated with ONX 0914 (Fig 8B). Since MAPkinases were indeed involved in TLR7-dependent cytokine induction and ONX 0914 negatively influenced TLR7-induced responses, we pursued the hypothesis that ONX 0914 affects MAPkinase activation upon TLR7 engagement. In fact, under the influence of ONX 0914, the abundance of the respective phosphorylated kinases p-p38, p-ERK1/2, and p-JNK was significantly reduced in TLR7 agonist-treated cells (Fig 8C and D). From these experiments, we conclude that PAMP-induced activation (ssRNA) of respective pattern recognition receptors (PRRs) (TLR7) and thereby affected MAPkinase signaling events in BMM are influenced by ONX 0914 inhibition.

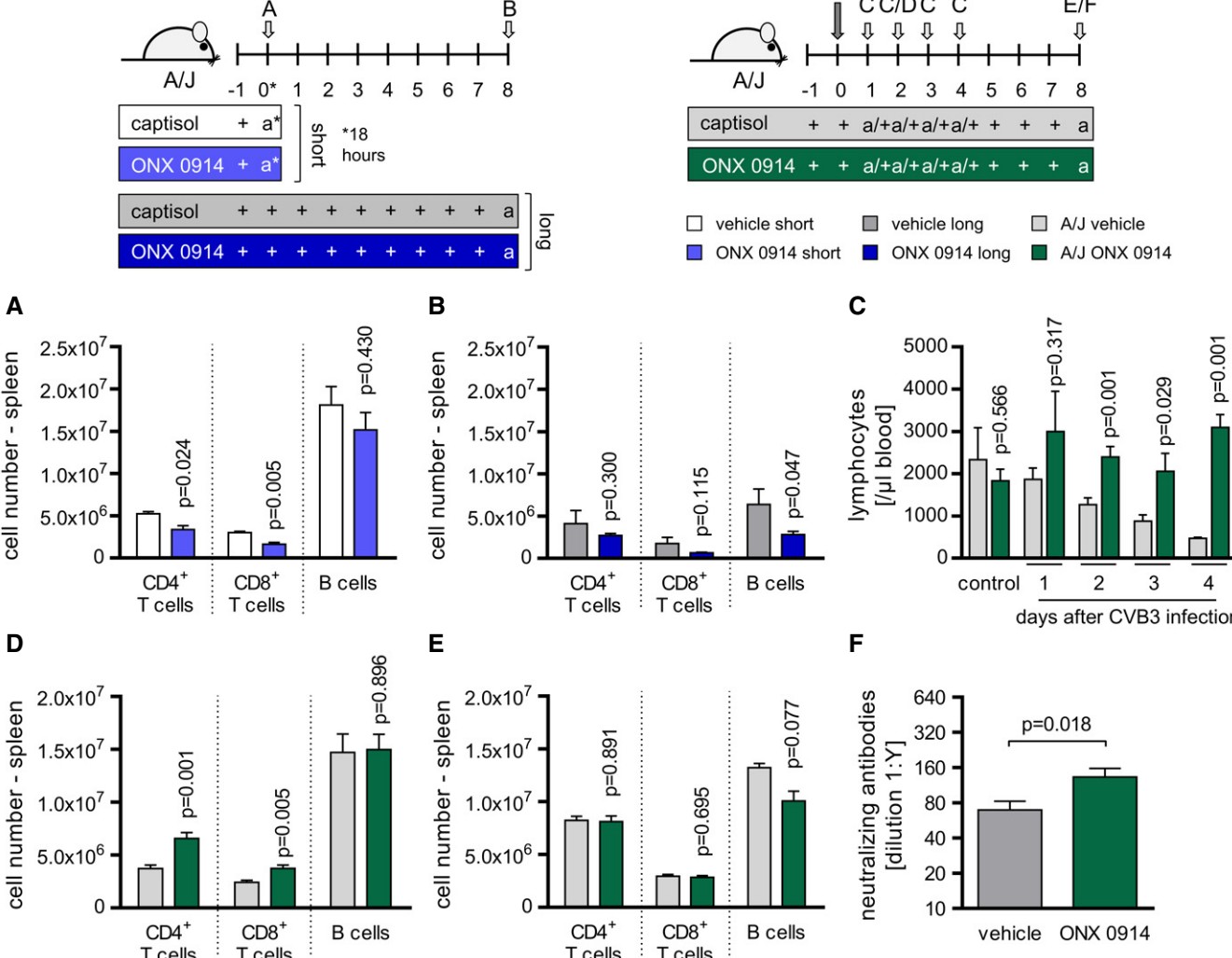

**Figure 4.  Abundance and distribution of lymphocytes under the influence of ONX 0914 and during CVB3 infection.**

A, B    Naive A/J mice were treated solely with ONX 0914 or vehicle for either a short time (18 h; light blue vs. white bars) (A) or long term (daily for 9 days; dark blue vs. gray bars) (B). Subsequently, spleen was isolated and analyzed using flow cytometry. Plotted are means of total numbers of the indicated lymphoid (sub-) populations + SEM (1–2 separate experiments were carried out; 18 h: vehicle $n$ = 2–3, ONX $n$ = 2–3; d8: vehicle $n$ = 4, ONX $n$ = 6). Unpaired $t$-tests were performed. $P$-values are indicated in each graph.

C–F    Starting one day prior to CVB3 infection, A/J mice were treated daily with either ONX 0914 or vehicle. At the indicated days after infection, mice were sacrificed, blood (C) (seven separate experiments were carried out; vehicle $n$ = 3–9, ONX $n$ = 3–7) and spleen (D, E) (1$_{(E)}$–2$_{(D)}$ separate experiments were carried out; d2: vehicle $n$ = 6, ONX $n$ = 6; d8: E3 vehicle $n$ = 3, ONX $n$ = 8) were isolated, and the number of lymphocytes in general or of the indicated lymphocyte subsets was determined by either automatic complete blood count analysis (C) or flow cytometry (D), respectively. CVB3-neutralizing antibody titers in the serum (F) were determined after 8 days (E3: vehicle $n$ = 10, ONX $n$ = 15). Means ± SEM are shown, and unpaired $t$-test was performed. $P$-values are indicated in each graph.

## Discussion

To improve clinical outcome and potentially prevent heart failure or even sudden death in hosts exhibiting high susceptibility to developing severe acute and chronic heart pathology upon pathogen encounter, new therapeutic approaches are in dire need. Predominantly severe inflammation of infected heart tissue is related to poor outcome in patients with suspected virus-induced myocarditis (Kindermann *et al*, 2008). Based on that, we considered an approach to antagonize adverse immune response

activation by inhibiting immunoproteasome function utilizing the ONX 0914 compound (Muchamuel *et al*, 2009). The present study revealed ONX 0914 treatment as utmost protective during virus-induced heart disease in mice leading to maintained survival and improved hemodynamic performance. The course of acute ONX 0914-influenced myocarditis was strongly attenuated and characterized by significantly reduced pro-inflammatory cytokine/chemokine expression, less infiltration, and reduced heart tissue damage. Our study underlines the substantial contribution of immunoproteasome proteolysis to stimulate an overwhelming myocardial

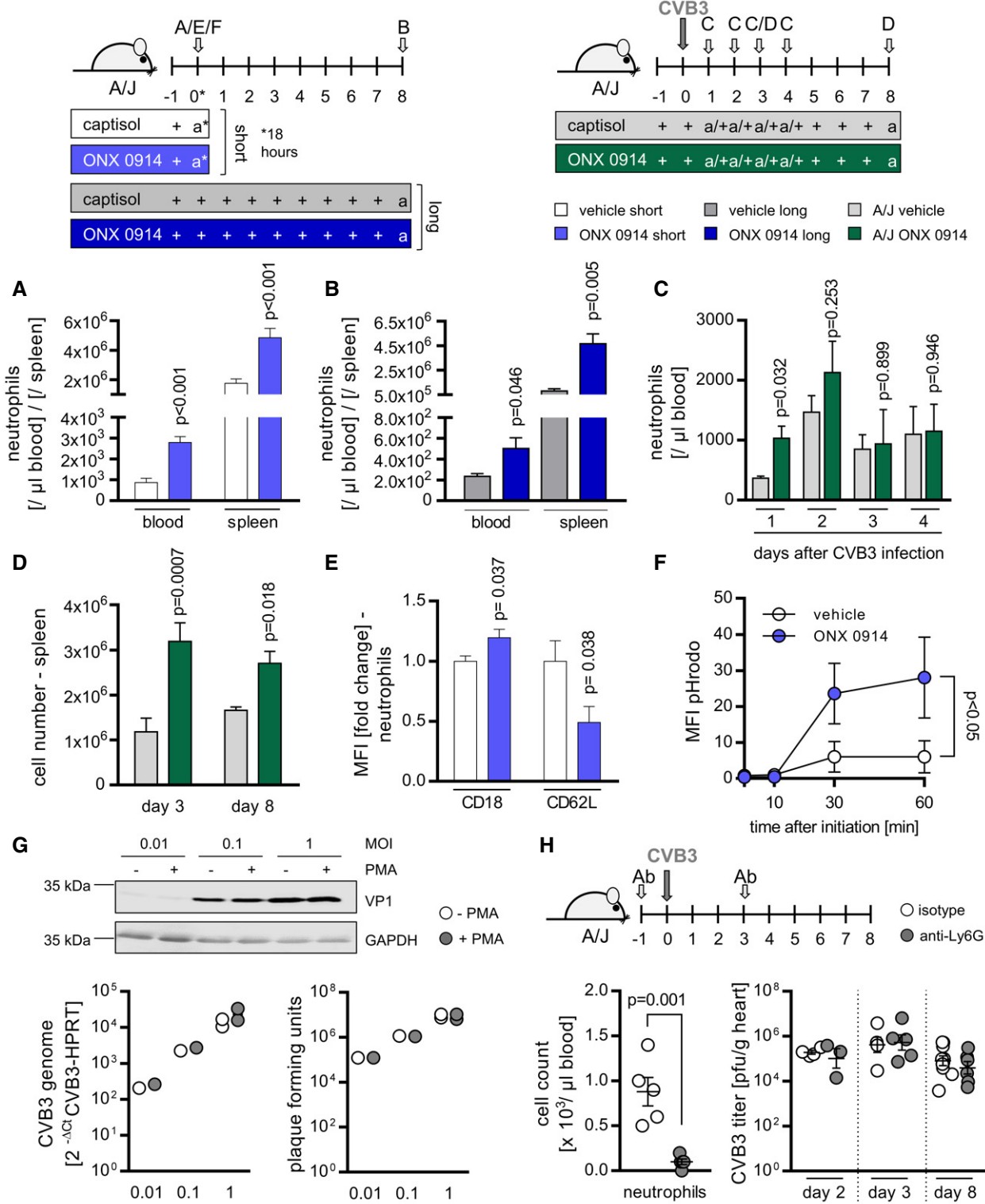

**Figure 5.**

immune response upon virus encounter eventually leading to sudden death and congestive heart failure. Based on this study and our previous work (Szalay *et al*, 2006; Jakel *et al*, 2009; Opitz *et al*, 2011), we suggest that in immune-genetically predisposed individuals, immunoproteasome activity is directly connected with adverse clinical outcome during virus-mediated inflammatory heart disease, and thus, immunoproteasome function represents a promising target for therapeutic intervention.

◀

**Figure 5.   Impact of ONX 0914-induced neutrophilia during CVB3 infection.**

A, B   Naive A/J mice were treated solely with ONX 0914 or vehicle for either a short time (18 h; light blue vs. white bars/dots) or long term (daily for 9 days; dark blue vs. gray bars). Subsequently, blood and spleen were isolated and analyzed for neutrophil abundance using flow cytometry. Plotted are means of total neutrophils + SEM. Unpaired *t*-tests (+ Welch correction) were performed (1 (B) or 7 (A) separate experiments were carried out; vehicle *n* = 4–10, ONX *n* = 4–7).

C, D   Next, A/J mice were infected with CVB3. ONX 0914 or vehicle treatment was carried out daily, starting one day prior to virus inoculation. At the indicated days after infection, mice were sacrificed, blood (C) (7 separate experiments were carried out; vehicle *n* = 3–9, ONX *n* = 3–7) and spleen (D) ($1_{(day\ 8)}$–$4_{(day\ 3)}$ separate experiments were carried out, respectively; d3: vehicle *n* = 9–12, ONX *n* = 10–12; d8: E3 vehicle *n* = 3, ONX *n* = 8) were isolated, and the number of different myeloid cell types was determined by either automatic complete blood count analysis or flow cytometry, respectively. Means + SEM are shown and unpaired *t*-tests (+ Welch correction) were performed.

E   After a single ONX 0914 injection into naive A/J mice, $CD11b^+/Ly6G^+$ splenic neutrophils were further characterized by assessing MFI for the surface markers CD18 and CD62L. Fold change of MFI relative to vehicle controls + SEM is shown, and unpaired *t*-tests were performed (two separate experiments were carried out; vehicle *n* = 7, ONX *n* = 7).

F   Splenocytes from ONX 0914- or vehicle-treated mice were first labeled with fluorochrome-conjugated antibodies against CD11b and Ly6G. To visualize and monitor phagocytosis, cells were incubated with fluorophore-labeled particles for the indicated time and analyzed using flow cytometry. After gating on $CD11b^+/Ly6G^+$ neutrophils, amount of phagocytosed-labeled particles was determined by assessing MFI per $10^5$ cells. Means ± SEM are depicted (one representative experiment out of two is shown; vehicle *n* = 4, ONX *n* = 4; two-way ANOVA followed by Bonferroni's multiple comparisons test).

G   Neutrophils were incubated with PMA to induce NETosis or DMSO as a control. Cells were simultaneously infected with CVB3 at the indicated MOIs. After 4 h, the supernatant was transferred onto HeLa cells. After 6 hours, expression of viral protein 1 (VP1) was analyzed by Western blotting, viral RNA was quantified by TaqMan qPCR, and infectious progeny formation was determined by a plaque reduction assay. Western blot: one representative experiment is demonstrated; a total of *n* = 1 for MOI 0.01 and MOI 0.1 as well as *n* = 2 separate experiments were conducted for MOI 1, respectively).

H   Male A/J mice were treated with anti-Ly6G or isotype control antibody one day prior to infection. For a day 8 infection experiment, antibody injection was repeated on day 3 p.i. and for a day 3 experiment on day 1 p.i. Mice were inoculated with a viral dose of $3.3 \times 10^3$ PFU to ensure survival of infected mice. Peripheral neutrophil abundance was determined by automatic blood counting after a total of two Ly6G injections (day −1 and day 1 p.i.) on day 3 p.i. Viral load in heart tissue was determined by standard plaque assay (one experiment for each time point (isotype *n* = 4, 5, 10, anti-Ly6G *n* = 3, 5, 6 for day 2, 3 and 8 respectively, *t*-test for cell count and two-way ANOVA for CVB3 titer)). Transformed means + SEM are presented.

Data information: *P*-values are indicated in each graph where applicable.
Source data are available online for this figure.

In contrast to former studies using germline LMP7-deficient mouse models, in which lack of immunoproteasome function is at least partially compensated by increased standard proteasome formation (Opitz *et al*, 2011), ONX-0914 treatment predominantly blocks LMP7-mediated protein hydrolysis (Muchamuel *et al*, 2009; Huber *et al*, 2012). Most of our knowledge on the biological value of the superior peptide hydrolysis capacity of the immunoproteasome in comparison with the standard proteasome originates from studies that had been conducted in LMP7-deficient mice being on a C57BL/6 background. Mice of this as resistant deemed genetic background (Chow *et al*, 1991) reflect a patient population, where no treatment would need to be initiated after pathogen encounter of the heart. In such mice, immunoproteasome formation especially after cytokine stimulus is known to be involved in maintaining protein homeostasis by degrading damaged proteins (Seifert *et al*, 2010; Opitz *et al*, 2011; Ebstein *et al*, 2013). This is in clear contrast to the picture observed in A/J mice exhibiting high susceptibility to virus-mediated inflammation (Chow *et al*, 1991; Opavsky *et al*, 2002). In this host, disturbances of immune homeostasis also involved imbalanced proteostasis as indicated by increased abundance of ubiquitinated conjugates in heart tissue sections (Fig EV5). Suppression of immunopathology achieved by immunoproteasome inhibition ultimately also influenced proteostasis in this host.

Another major physiologically relevant function of immunoproteasomes found during viral infection is a more efficient generation of viral peptides resulting in improved antigen presentation of MHC class I epitopes (Schwarz *et al*, 2000; Kincaid *et al*, 2012). Although facilitated antigen processing by the immunoproteasome is also evident for coxsackievirus peptides *in vitro* (Jakel *et al*, 2009; Voigt *et al*, 2010; Respondek *et al*, 2017), this immunoproteasome-dependent improvement of epitope liberation has no effect on the course of CVB3 infection *in vivo* (Opitz *et al*, 2011). Interestingly as shown in this study, ONX 0914 treatment in C57BL/6 mice with hereditary resistance to viral cardiomyopathy slightly deteriorated disease

parameters like viral load without affecting overall long-term course. Such effects could be at least partially attributed to the solid T1IFN response this host induces to combat CVB3 infection (Jakel *et al*, 2009; Rahnefeld *et al*, 2011), and which was significantly reduced after ONX 0914 application being in line with previous findings by others (Muchamuel *et al*, 2009; Ichikawa *et al*, 2012). Importantly, in C57BL/6 mice, no significant impact on induction of an efficient specific immune response regarding formation of CVB3-neutralizing antibodies could be observed. Intact activation of adaptive immunity with firm memory protection from viral heart disease represents important safety prerequisites of ONX 0914 regarding its putative clinical application.

Mouse models representing high susceptibility to viral myocarditis demonstrated a shifted and overall significantly impaired T1IFN response (Fig EV3; Jakel *et al*, 2009), and in case of A/J mice used in this study, increased mortality and sepsis-like disease course of CVB3 infection. The question arose how ONX 0914 treatment exerts the demonstrated remarkably beneficial effect on disease and even on mortality in this host. In addition to exocrine pancreatic tissue, cardiotropic viruses like CVB3 directly destroy cardiomyocytes, and this virus-mediated tissue destruction stimulates immune cell infiltration (Corsten *et al*, 2012; Rahnefeld *et al*, 2014). Particularly, the recognition of viral pathogen motifs—like single-stranded RNA in the case of CVB3—by the host's PRR initiates the expression of inflammatory cytokines/chemokines (Mann, 2011). Since we observed substantial reduction in these immune active molecules under ONX 0914 influence in A/J mice, we analyzed whether an altered myocardial viral burden might be involved in this decreased heart tissue inflammation. The local viral burden was slightly reduced 2 days after infection in the heart. However, with ongoing infection myocardial levels of infectious virus particles of both treatment groups converged. Yet, ONX 0914-induced immune-modulation was pronounced throughout disease course and particularly significant even though cardiac pathology was fully defined 8 days

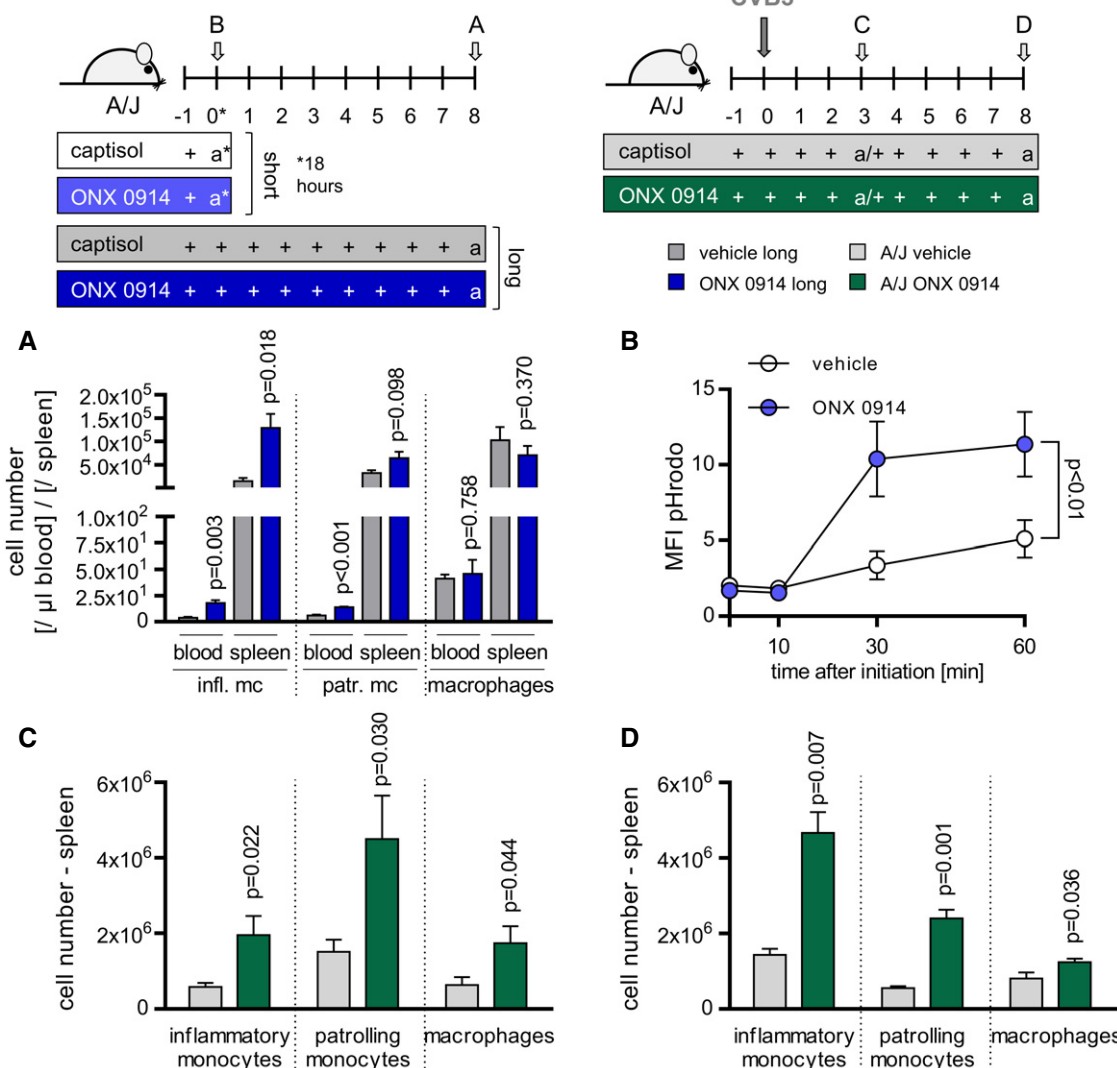

**Figure 6.  Impact of ONX 0914 on abundance and dissemination of monocytes/macrophages during CVB3 infection.**

A, B   Naive A/J mice were treated solely with ONX 0914 or vehicle daily for 9 days (A) (dark blue vs. gray bars) or with a single shot (B) (18 h; light blue vs. white bars/dots). Subsequently, blood and spleen (A) were isolated and analyzed using flow cytometry. Plotted are means of total numbers of the indicated myeloid (sub-)populations + SEM. Unpaired *t*-tests (+ Welch correction) were performed (one experiment; vehicle *n* = 4, ONX *n* = 6). Splenocytes from ONX 0914- or vehicle-treated mice were first labeled with fluorochrome-conjugated antibodies against the surface markers CD11b and F4/80 (B). To visualize and monitor phagocytosis, cells were incubated with fluorophore-labeled particles for the indicated time and analyzed using flow cytometry. After gating CD11b$^+$/F4/80$^+$ macrophages, amount of phagocytosed labeled particles was determined by MFI per $10^5$ cells (means + SEM) (one representative experiment out of two is shown; vehicle *n* = 4, ONX *n* = 4; two-way ANOVA followed by Bonferroni's multiple comparisons test). *P*-values are indicated in each graph.

C, D   A/J mice were infected with CVB3. ONX 0914 or vehicle treatment was carried out daily, starting one day prior to virus inoculation. 3 (C) and 8 days (D) after infection, mice were sacrificed, spleen was isolated (1 (D) or 4 (C) separate experiments were carried out, respectively; d3: vehicle *n* = 9–12, ONX *n* = 10–12; d8: E3 vehicle *n* = 3, ONX *n* = 8), and the number of different myeloid cell types was determined by flow cytometry. A detailed description of the gating strategy for identification of the different immune cell type is provided in Fig EV2. Means + SEM are shown, and unpaired *t*-tests (+ Welch correction) were performed. *P*-values are indicated in each graph.

after virus inoculation. Therefore, we concluded that ONX 0914-induced impact on cytokine production is independent from the establishment of virus infection in a particular organ and most likely involves other mechanisms.

Since the unimpaired effectiveness of innate immune mechanisms and the closely connected resulting adaptive immune status of the host are immensely crucial for disease outcome (Epelman *et al*,

2015), we characterized the existing immune status during CVB3 infection and the potential immune-modulating influence, ONX 0914 might exert, explaining the improved signs of disease. Being present already prior as well as during virus infection, ONX 0914 directly influenced neutrophils, which are the first and most abundant cell population of the host's innate immune response (Jenne & Kubes, 2015). Not only abundance was significantly increased in blood and

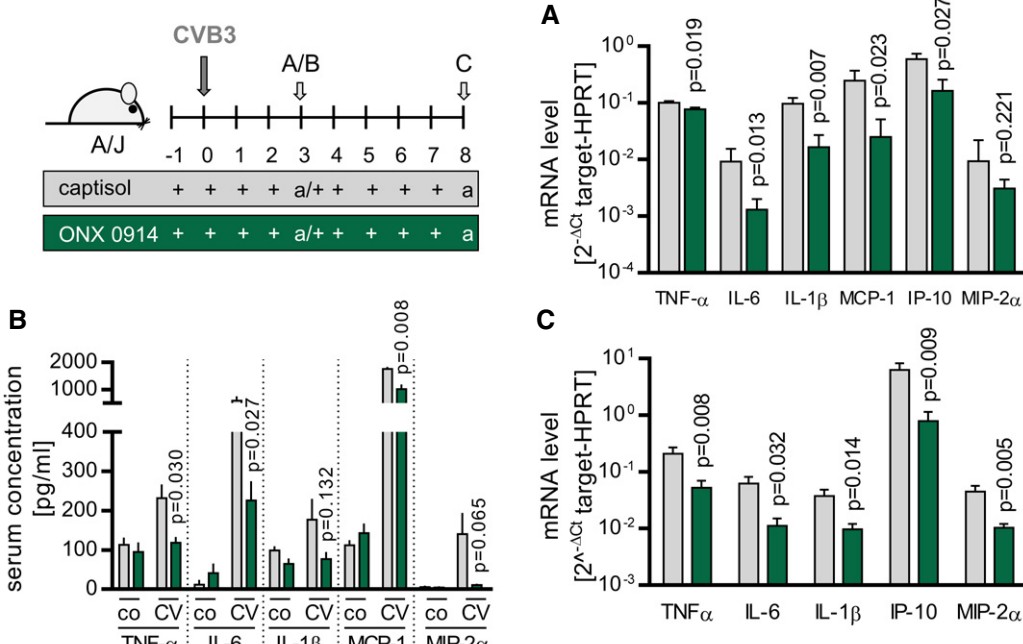

**Figure 7.  ONX 0914-mediated suppression of CVB3-induced pro-inflammatory cytokines/chemokines in A/J mice.**

A–C  A/J mice were infected with CVB3. ONX 0914 or vehicle treatment was carried out daily, starting one day prior to virus inoculation. Animals were sacrificed after 3-day (A, B) and 8-day p.i. (C). Total spleen mRNA (A) was used to determine cytokine/chemokine-specific expression of the indicated genes by TaqMan qPCR. Means of $2^{-\Delta Ct}$ + SEM are shown (vehicle $n = 7$, ONX $n = 9$, $t$-tests). Serum concentration of the indicated cytokines/chemokines (B) was assessed using bead-based multiplex immunoassay or ELISA, respectively. Depicted are means + SEM (four separate experiments were carried out; vehicle $n = 3–7$, ONX $n = 3–7$, $t$-tests). During the peak of heart muscle inflammation, total heart mRNA (C) was used to determine cytokine/chemokine-specific expression of the indicated genes by TaqMan qPCR. Means of $2^{-\Delta Ct}$ + SEM are shown (E1 + E2; vehicle $n = 4$, ONX $n = 12$, $t$-tests). $P$-values are indicated in each graph.

in spleen. Neutrophils were also set into an advanced activated state. Although the function of neutrophils in antiviral defense is not fully resolved, several publications reported neutrophil recruitment in virus infection being an actual part of a protective strategy, orchestrated by the innate immune system leading to protective defense against viral disease (Jenne & Kubes, 2015). Therefore, it seemed likely that ONX 0914-induced neutrophilia could also influence CVB3. Nevertheless, NET formation, which is considered a powerful property of neutrophils to combat also viruses (Saitoh *et al*, 2012; Jenne *et al*, 2013), did not influence CVB3. More supportive evidence on the biological function of neutrophils in the battle against CVB3 was obtained *in vivo*. Since neutrophil depletion had no significant impact on CVB3-induced pathology, we concluded—that consistent with a previous report on unaltered CVB3-induced pathology in granulocyte-colony-stimulating factor (G-CSF)-treated mice (Hiraoka *et al*, 1995)—ONX 0914-induced neutrophilia most likely does not majorly attribute to diminished immunopathology in A/J mice. No significant difference was seen in the survival, cardiac disease, or myocardial virus titers between the G-CSF and the control groups. Nevertheless, the impressive mobilization and functional activation of neutrophils which we achieved by ONX 0914 treatment independent of pathogen encounter are of high relevance for bacteria-induced inflammatory syndromes with significant biological impact of neutrophils.

In addition to neutrophils, ONX 0914 treatment also mobilized monocytes/macrophages from the bone marrow leading to increased numbers in blood and spleen. On the one hand, Ly6C[high] monocytes are equipped with migration ability and represent immediate circulating precursors for antigen-presenting dendritic cells (Ginhoux & Jung, 2014). Having in mind that ONX 0914-treated A/J mice show higher titers of CVB3-neutralizing antibodies, one might argue that the significantly elevated numbers of those antigen-presenting dendritic cell precursors could be as well involved in a more effective onset of such an antiviral immune response. On the other hand and decisive in our context, Ly6C[high] monocytes can give rise to macrophages under inflammatory conditions (Ginhoux & Jung, 2014). In clear contrary to the observed enrichment of Ly6C[high] monocytes in spleens of ONX-treated A/J mice, acute myocardial injury as reflected by the extent of monocyte/macrophage infiltration and accompanying pro-inflammatory cytokine response activation (Corsten *et al*, 2012) was significantly reduced under ONX 0914 influence. This is of utmost importance in this context since the peripheral abundance, for example, of inflammatory monocytes can actually contribute to severe cardiac immunopathology (Leuschner *et al*, 2015). Mononuclear phagocytes are responsible for removal of cell remnants originating from infected and dying cells. Therefore, a low abundant population might be beneficial for eliminating and resolving inflammation (Corsten *et al*, 2012; Epelman *et al*, 2015). In fact, we propose that immunoproteasome inhibitor treatment might even facilitate this important function of macrophages since we could clearly demonstrate a directly ONX 0914-mediated increase in phagocytic capacity exerted by these cells.

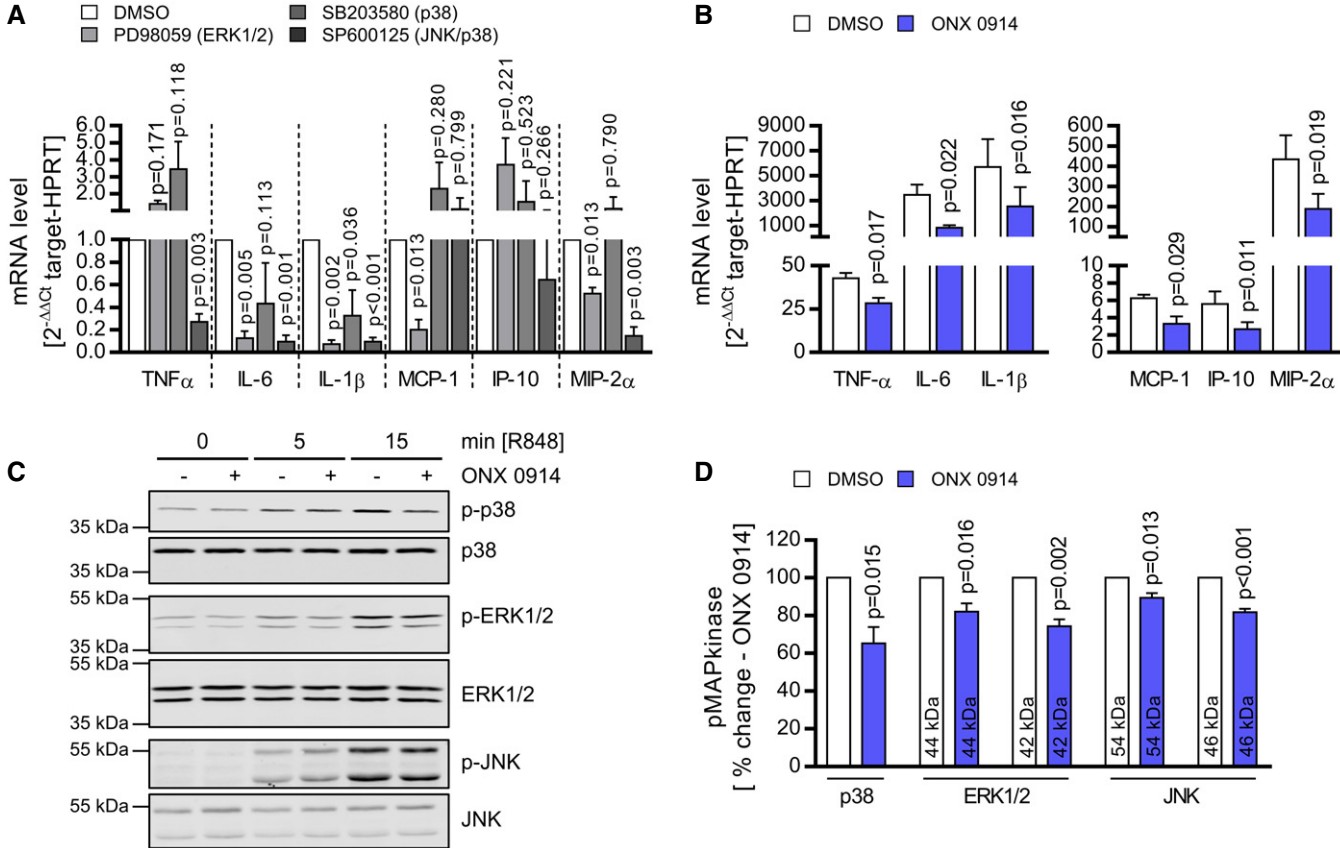

**Figure 8. ONX 0914 manipulates TLR7 signaling leading to reduced MAPkinase phosphorylation.**

A    Bone marrow-derived macrophages (BMM) from A/J mice were cultivated. Mitogen-activated protein kinase (MAPkinase) inhibitors were used to investigate the contribution of p38-, ERK1/2-, and JNK-mediated signaling to cytokine/chemokine expression upon TLR7 engagement. Therefore, BMM were treated with the MAPkinase inhibitors SB203580 (p38), PD98059 (ERK1/2), SP600125 (JNK), or DMSO (control), respectively, for one hour prior to R848 stimulation. After 8 h, mRNA levels of the indicated genes were determined by TaqMan qPCR. Data represent fold change increased mRNA expression ($\Delta\Delta C_t$ normalized to stimulated DMSO control) and are mean of three independent experiments. Paired *t*-tests (inhibitor versus DMSO) were performed.

B    BMM from A/J mice were treated with ONX 0914 or DMSO prior to stimulation with the TLR7 agonist R848. After 8 h, cytokine and chemokine expression was determined for the indicated genes by TaqMan qPCR. Data represent fold change increased mRNA expression (normalized to un-stimulated DMSO control) and are mean of four independent experiments. Paired *t*-tests were performed.

C, D   BMM from A/J mice were treated with ONX 0914 or DMSO prior to stimulation with R848 for the indicated points in time. Cytosolic protein extracts were subjected to Western blot analysis for detection of the phosphorylation status of p38 (Thr180/Tyr182), ERK1/2 (Thr202/Tyr204) and JNK (Thr183/Tyr185) (C). Antibodies directed against the respective total MAPkinases were used as respective loading control. The most intense phosphorylation status for all MAPkinases was observed after 15-min TLR7 engagement. Signal intensity at this point in time was determined for DMSO- and ONX 0914-treated cells (D), and signals of phosphorylated kinases were normalized to the respective non-phosphorylated protein. For three independent experiments, thereby obtained data of ONX 0914-treated cells were normalized to respective DMSO controls and plotted bar graphs demonstrate mean of the respective phosphorylated MAPkinases (unpaired *t*-test).

Data information: Data presented as mean ± SEM. *P*-values are indicated in each graph.
Source data are available online for this figure.

Nevertheless, the magnitude of immune response activation in infected hearts determines disease outcome (Kindermann *et al*, 2008). Monocytes/macrophages do not only limit tissue injury, but they are also increasing acute inflammatory and fibrotic tissue damage (Hirasawa *et al*, 1996; Jaquenod De Giusti *et al*, 2015). The main effector molecules that influence the migration and local acute as well as chronic fibrotic injury are chemokines and pro-inflammatory cytokines, respectively. In fact, for most of the ONX 0914-influenced cytokines such as TNF-α, IL-6, and IL-1 previous studies have convincingly demonstrated that their concentration as well as the timing of their release determine how these immune-modulatory

molecules actually affect the outcome of virus-induced myocarditis, whereby high cytokine levels result in more severe pathology (Corsten *et al*, 2012). According to that, clinical studies reported on the adverse effects of these mediators in humans suffering septic shock revealing both a separate as well as synergistic cardio-depressive effect of TNF-α and IL-1 (Kumar *et al*, 1996; Cain *et al*, 1999). Therefore, the ONX 0914-mediated suppression of cytokine production observed in this study is most likely a driving force for improvement of cardiac condition and improved survival in A/J mice. Moreover, ONX 0914 treatment of predisposed hosts during acute state of virus infection might also beneficially affect long-term

disease outcome since the extent of myeloid cell infiltration and cytokine production also substantially influences the activation of cardiac remodeling and chronic pathology (Corsten *et al*, 2012; Rahnefeld *et al*, 2014; Epelman *et al*, 2015; Leuschner *et al*, 2015).

In previous attempts to profile the role of immunoproteasome proteolysis for pro-inflammatory effector cascades, others and we have reported on modified effector responses upon engagement of TLRs, T-cell receptor, and TNF-α receptor, yet the underlying pathways were less well characterized. Nevertheless, there is consensus that ONX 0914 does not directly influence canonical NFκB signaling (Muchamuel *et al*, 2009; Paeschke *et al*, 2016; Bitzer *et al*, 2017). Other than that and in line with TLR4-mediated pentraxin3 production (Paeschke *et al*, 2016), we provided compelling evidence for reduced ERK1/2, p38, and JNK phosphorylation in TLR7-activated BMM upon ONX 0914 treatment. Others and we confirmed that therein-affected downstream effector kinase activity is as well involved in pro-inflammatory cytokine/chemokine production (Ronkina *et al*, 2007). Hence, thereby affected pathways might also contribute to the impressively reduced local and systemic cytokine response we found under ONX 0914 influence in spleen, serum, and heart tissue in A/J mice. This finding is even more impressive since the abundance of monocytes/macrophages was increased in spleen by ONX 0914 at the same point in time. Our previous studies have demonstrated that, in comparison with its standard proteasome counterpart, the immunoproteasome is equipped with a superior peptide hydrolysis capacity (Jakel *et al*, 2009; Voigt *et al*, 2010; Mishto *et al*, 2014). Hence, we propose that immunoproteasome proteolysis modulates the abundance and/or activity of certain kinases, phosphatases and/or regulatory proteins that are involved in, for example, the complex MAPkinase signaling network. To identify such affected proteins, a comprehensive system biology-based approach is needed in the future.

Taken together, ONX 0914 potently inversed high susceptibility to virus-mediated acute myocarditis primarily based on a local and systemic suppression of pro-inflammatory cytokine/chemokine production and resulting detrimental conditions. While destructive immune responses were reduced, pathogen-directed immunity was maintained or considerably improved. Resulting maintenance of immune homeostasis reversed inflammatory heart tissue damage and, most strikingly, preserved cardiac output and survival upon pathogen encounter. These immune-modulatory features exerted by ONX 0914 during the course of viral myocarditis kept the cardiac output close to baseline level. Consequently, immunoproteasome-specific inhibitors like ONX 0914 rank among novel drugs for the treatment of severe acute myocarditis. Moreover, the utmost beneficial immune-modulatory impact found upon immunoproteasome inhibitor treatment of infected mice warrants further preclinical testing of such compounds for systemic inflammatory response syndromes and sepsis.

# Materials and Methods

### Animals, virus, and infection protocols

Original mating pairs of the two different wild-type mouse strains—A/J and C57BL/6—were purchased from Harlan Winkelmann (A/J) and Jackson Laboratory (C57BL/6). For E3 day 8 and neutrophils

depletion studies conducted in A/J mice, larger cohorts were purchased from Harlan Winkelmann and allowed to settle for at least 1 week prior to virus inoculation. With the exception of E1 + E2 (mixed gender was used), male adult in-house bred animals (6- to 8-week-old) were injected intraperitoneally (i.p.) with $10^4$ PFU (A/J) and $10^5$ PFU (C57BL/6) of a cardiotropic variant of CVB3 Nancy strain (Klingel *et al*, 1992). Average weight of A/J mice included in experiment E1 and E2 was vehicle 20.4 g ± 0.8; ONX 20.6 g ± 0.93. As a large percentage of vehicle-treated mice succumbed to infection, for E3, slightly older and consequently heavier mice were used: vehicle 23.0 g ± 0.3; ONX 23.7 g ± 0.4. To analyze CVB3-induced immune memory status, re-infection experiments were carried out in C57BL/6 mice. Therefore, mice were initially inoculated with $10^6$ PFU CVB3 and 28-day p.i. challenged a second time with $10^5$ PFU CVB3. For neutrophil depletion, mice were injected i.p. with 50 μg low-endotoxin acid-free (LEAF) Ly6G antibody (clone 1A8) or respective IgG2a isotype control (BioLegend) one day prior to infection. Antibody delivery was repeated after 48 h for mice that were sacrificed 3-day p.i. or after 72 h for mice that were sacrificed 8-day p.i. This study was carried out in accordance with the recommendations in the Guide for the Care and Use of Laboratory Animals of the German animal welfare act, which is based on the Directive of the European Parliament and of the Council on the protection of animals used for scientific purposes. The protocol was approved by the Committee on the Ethics of Animal Experiments of Berlin State authorities (G0274/13). All efforts were made to minimize suffering.

### Echocardiography

For echocardiography, mice were anesthetized with 1.5–2% isoflurane and kept warm on a heated platform. Temperature and ECG were continuously monitored. Cardiac function and morphology were assessed with a VisualSonics Vevo 770 High-Frequency Imaging System using a high-resolution (RMV-707B; 15–45 MHz) transducer. Standard imaging planes, M-mode, and functional calculations were obtained. Pulsed wave Doppler measurements were acquired from a modified apical two-chamber view on the mitral valve. The parasternal long-axis view of the left ventricle (LV) was used to guide calculations of percentage fractional shortening, ventricular dimensions and volumes. M-mode echocardiographic images were recorded at the level of the papillary muscles from the parasternal short-axis view. All measurements were performed by an experienced, blinded technician.

### Cell culture

Cells of peripheral blood (PBC) were purified from whole blood, either obtained from puncture of the Vena facialis or by heart puncture. RBC lysis was performed 2–3 times in a row by incubating with 0.83% ammonium chloride ($NH_4Cl$) for 3–5 min at room temperature. Splenocytes (SPC) were prepared by passing spleen tissue through a 70-μm cell strainer (BD Bioscience). After a wash step with 1× PBS, RBC were lysed in one step as described for blood cells. All cell types were recovered by centrifugation (10 min, 310 *g*), resuspended in FACS buffer, and chilled on ice until flow cytometry.

Mouse bone marrow cells were isolated from A/J mice and cultivated in the presence of RPMI medium supplemented with 30%

L929 cell-conditioned medium as a source of granulocyte/macrophage colony-stimulating factor as previously described (Paeschke et al, 2016). Cells were treated with 100 nM ONX 0914 or DMSO (Paeschke et al, 2016), respectively, prior to stimulation with the TLR7/8 agonist R848 (10 µg/ml, StemCell Technologies. Alternatively, bone marrow-derived macrophages were treated with 75 µM PD98059 (ERK1/2), 10 µM SB203580 (p38), and 25 µM SP600125 (JNK) [all compounds were purchased from Invivogen] for 1 h prior to TLR7 activation.

Neutrophils were isolated from human peripheral blood according to procedures outlined elsewhere (Brinkmann et al, 2010). Briefly, neutrophils were separated from whole blood by density centrifugation using Histopaque-1119 (Sigma-Aldrich) followed by a Percoll-gradient centrifugation (Santa Cruz Biotechnology). $5 \times 10^5$ neutrophils were seeded in RPMI medium supplemented with 2% human serum albumin (Grifols), and cells were incubated at 37°C and 5% $CO_2$ for 1 h. Formation of neutrophil extracellular traps (NETs) was induced by 100 nM phorbol 12-myristate 13-acetate (PMA, Calbiochem). NET formation was confirmed microscopically by Hoechst staining after 4 h. In parallel, cells were infected with CVB3 Nancy at MOI 0.01, 0.1, and 1.0. After 4 h, supernatants were transferred onto confluent HeLa cells. Cells were incubated for 30 min at 37°C with 10-fold dilutions of CVB3-containing supernatants and directly overlaid with agar-containing MEM. Two to three days later, cells were stained with 0.025% neutral red/PBS and plaque-forming units (PFU) were counted 3 h after staining. Furthermore, HeLa cells were inoculated with virus containing supernatants and cells were washed after 1 h with PBS. Six hours after inoculation, RNA and protein were isolated, viral RNA was quantified by TaqMan qPCR, and viral protein production was determined by VP1 Western blotting.

For in cellulo infection studies, primary embryonic cardiomyocytes (CM) were obtained from C57BL/6 mice on embryonic day 14 and cultured as described elsewhere (Spur et al, 2016). ONX 0914 or DMSO was applied to each well, and simultaneously cells were infected with CVB3 at MOI 5. Infectious virus particles were allowed to settle and attach to the cell surface for 1 h. Afterward, unbound virions were removed and cell culture was supplied with virus-free but ONX-0914- or DMSO-containing medium. Another 2 h later, medium was aspirated once more and replaced by inhibitor-/DMSO-free DMEM. At the indicated times p.i., cell culture wells were quick-frozen, thawed again, and clear supernatant was analyzed for concentration of infectious virus particles by $TCID_{50}$ assay.

**Proteasome inhibitor and treatment protocols**

ONX 0914 is a cell-permeable β5i-/LMP7-selective inhibitor (ONYX/ Amgen, San Francisco, CA or Selleckchem, San Diego, CA). ONX 0914 was formulated in an aqueous solution of 10% (w/v) Captisol™ (Ligand Pharmaceuticals SanDiego, CA) and 10 mM sodium citrate (pH 6). The solution was subcutaneously (s.c.) administered to mice at a dose of 10 mg/kg in a volume of 200 µl. Control group animals received matching amounts of Captisol–sodium citrate mix (referred to as vehicle).

**Histological images and analysis**

Heart tissue of infected and differentially treated mice was isolated and immediately fixed in Histofix (1× PBS, 4% Roth™ Histofix)

over-night. Afterward, Histofix was replaced by 1 × PBS, tissue was paraffin embedded, and sections were prepared. Histological staining with hematoxylin/eosin (H&E) or Masson's trichrome staining as well as immunohistochemistry for the detection of $CD3^+$ T lymphocytes and $Mac3^+$ macrophages was carried out as described elsewhere (Kindermann et al, 2008). Area of inflammation/$Mac3^+$-positive area of inflammation was assessed as described in reference (Rahnefeld et al, 2014). ImageJ software was used.

**Isolation of infiltrated immune cells from heart tissue**

After extraction of whole blood, heart was flushed with 15 ml PBS, removed, and washed again in PBS. An in terms of weight defined amount of heart tissue was minced in RPMI 1640 medium (Biochrom) containing 10% (v/v) fetal calf serum (FCS) (Biochrom), 1% (v/v) penicillin/streptomycin (Pan Biotech), 30 mM HEPES, 0.1% (w/v) collagenase type 2 (Worthington), and 0.015% (w/v) DNase I (Sigma-Aldrich). Tissue digestion occurred during incubation at 37°C for 30 min (shaking at 800 rpm). In order to obtain single-cell status, 10 mM EDTA was added, and cells were washed with PBS and were passed through a 70-µm cell strainer (BD Bioscience). Cells were recovered by centrifugation (10 min, 310 g), re-suspended in FACS buffer, and chilled on ice until flow cytometry.

**Flow cytometry**

Either equal numbers of cells (BMC, PBC, and SPC) or cells purified from 15 mg heart tissue were incubated (20 min at 4°C) in FACS buffer (phosphate-buffered saline, 2% FCS, 2 mM EDTA) containing an anti-mouse Fc receptor blocking reagent (1:50; Miltenyi). Afterward, fluorochrome-conjugated antibodies against various surface markers were added and incubated for at least 20 min at 4°C protected from light. The following anti-mouse antibodies were purchased from BD Bioscience: B220 (PE; clone RA3-6B2) (1:300), CD90.2/Thy-1.2 (PE; clone 53-2.1) (1:300), NK-T/NK Cell Antigen (PE; clone U5A2-13) (1:300), TER-119 (PE; clone TER-119) (1:300), CD11b (PE-CF594; clone M1/70) (1:300), CD4 (V500; clone RM4-5) (1:100), CD8α (Pacific Blue™; clone 53.6.7) (1:100), CD62L (PerCP Cy5.5; clone MEL-14) (1:300). Anti-mouse CD49b (PE; clone DX5) (1:300) was purchased from eBioscience/ Affymetrix, and the following anti-mouse antibodies were obtained from BioLegend: CD45.2 (Brilliant Violet 711™; clone 104) (1:200), Ly6G (PerCP/Cy5.5; clone 1A8) (1:400 or 1:600), Ly6C (Pacific Blue™; clone HK1.4) (1:200), CD11c (Brilliant Violet 510™; clone N418) (1:150), F4/80 (APC; clone BM8) (1:100), CD3 (PerCP/ Cy5.5; clone 145-2C11) (1:300), B220 (FITC; clone RA3-6B2) (1:200), CD19 (APC; clone 6D5) (1:500), CD18 (FITC; clone M18/ 2) (1:250). Anti-mouse CD4 (FITC; clone RM4-5) (1:200) was purchased from ThermoFisher Scientific. After several wash steps with FACS buffer (centrifugation: 3 min at 306 g), cells were re-suspended in 200 µl of the Fixable Viability Dye eFluor 780 (eBioscience), diluted 1:1,000 in PBS, and incubated for 30 min on ice protected from light. After serial wash steps with PBS followed by fixation in FACSFix (1× PBS, 2% Roth™Histofix), cells were acquired either on an LSRII or a FACSCalibur (both BD Bioscience). Data were analyzed using FlowJo v10.0 software (Tree Star). In order to quantify total cell numbers in heart tissue,

123count eBeads (eBioscience) were used according to manufacturer's protocol. Reported numbers were normalized for the weight of total hearts, yielding the number of respective cell fraction per mg tissue.

## ELISA and multiplex assay

IL-6 concentration in serum was determined using OptiEIA mouse IL6 enzyme-linked immunosorbent assay (ELISA) Set purchased from BD Biosciences. Levels of tumor necrosis factor α (TNF-α), interleukin-1β (IL-1β), monocyte chemoattractant protein-1 (MCP-1), and macrophage inflammatory protein (MIP-2) were assessed by bead-based multiplex immunoassay using Bio-Plex Pro™ Mouse Cytokine Standard Plex (Bio-Rad) performed according to the protocols provided by the manufacturer. IFN-β concentration in serum was determined using a Verikine™ Mouse IFN-β ELISA Kit purchased from PBL Assay Science.

## RNA isolation and quantitative real-time PCR

RNA isolation, cDNA synthesis, and quantitative real-time PCR were performed as described in publication (Paeschke *et al*, 2016). Primers and probes for TaqMan PCR were purchased from ThermoFisher (TaqMan™ Gene Expression Assays). mRNA expression was normalized to the housekeeping gene hypoxanthine-guanine phosphoribosyltransferase (HPRT) according to the $\Delta C_t$ or $\Delta\Delta C_t$ method.

## Western blot analysis and antibodies

For analysis of Toll-like receptor 7 (TLR7) signaling, ONX 0914-treated bone marrow-derived macrophages (100 nM, 16 h) or DMSO-treated cells were followed for 5, 15, and 30 min. Cellular protein extracts were separated from nuclear extracts according to the manufacturer's protocol of an Active Motif™ kit. Primary antibodies used were as follows: phospho-p38 (Thr180/Tyr182) (1:1,000), p38 (1:1,000), phospho-JNK (Thr183/Tyr185) (1:1,000), JNK (1:1,000), phospho-ERK1/2 (Thr202/Tyr204) (1:1,000), ERK1/2 (1:1,000) (all Cell Signaling). The bound primary antibodies were detected using IRDye800CW labeled goat anti-mouse (p-p38) (1:10,000)/anti-rabbit (all remaining) (1:10,000) secondary antibodies in conjunction with an Odyssey CLx infrared imaging system (Li-Cor Biosciences).

## Titration of virus in mouse tissue and neutralization test

The titer of infectious virus was determined by standard plaque assays, performed on subconfluent GMK cell monolayers as previously described (Rahnefeld *et al*, 2014). To determine the relative amount of CVB3-neutralizing antibodies, serum was diluted initially 1:10 and further serially 1:2 using DMEM. In quadruplicate, the different dilutions were mixed with an equal volume of medium containing 100 $TCID_{50}$ of CVB3 and were incubated for 1 h at 37°C, 95% humidified atmosphere, and 5% $CO_2$. Afterward, 100 μl GMK cell suspension was added to each dilution parallel and the test was incubated for at least 4 days at 37°C/5% $CO_2$. The dilution of serum, which protected cells from virus infection completely, was set as antibody titer.

## The paper explained

### Problem

In genetically predisposed patients, viral infection of the heart muscle can activate a devastating immune response, resulting in severe functional impairment of the heart, chronic organ failure, or even sudden death. We aimed to define a novel approach that targets adverse immune response activation, but ensures efficient pathogen clearance.

### Results

We utilized a compound that specifically targets a major catalytically active subunit called LMP7 of a multicatalytic protease complex—the immunoproteasome—virtually found in all human immune cells. ONX 0914 inhibits protein cleavage mediated by this subunit orthologue in mice. When we treated mice with high hereditary susceptibility to virus-mediated inflammation of the heart, we observed fundamental beneficial effects. Administration of ONX 0914 antagonized detrimental immune response activation and efficiently suppressed the pro-inflammatory cytokine storm that was characteristic and decisive for mortality and cardiac dysfunction of vehicle-treated mice. Compound-treated mice did not only survive, and they were in overall good condition and demonstrated substantially reduced signs of inflammatory heart tissue damage. We provided some molecular insights into how immunoproteasome proteolysis might influence a complex signaling immune network leading to overwhelming inflammation. Strikingly, the inhibition of the immune cell resident protease ensured pathogen elimination and proved to be safe also after long-term application.

### Impact

Immunoproteasome-specific inhibitors like ONX 0914 rank among novel drugs for the treatment of severe virus-mediated inflammation of the heart. The utmost beneficial immune-modulatory impact found upon compound treatment during infection warrants further testing for other sterile or pathogen-induced systemic inflammatory response syndromes.

## Phagocytosis assay

The ability of splenic neutrophils and macrophages to phagocyte was analyzed using pHrodo™ Green *Escherichia coli* BioParticles™ Conjugate for Phagocytosis (ThermoFisher) according to the manufacturer's instructions. To be able to gate on $CD11b^+/Ly6G^+$ or $F4/80^+$ populations, prior to administration, splenocytes were surface-labeled as described. While probes were incubated for the indicated time at 37°C/5% $CO_2$, negative controls rested on ice. Phagocytosis was stopped by placing the cells on ice. After one wash step, cells were immediately analyzed by flow cytometry using a FACSCalibur (BD Bioscience).

## Statistics

Statistical analysis of the data was performed in GraphPad Prism v6.00 or v7.00 for Windows (GraphPad Software, La Jolla, California, USA). Data are given as mean ±/+ standard error of the mean (SEM) unless specified otherwise. Logarithmic data (virus titer, semi-quantitative RNA quantification) measured on a linear scale were transformed logarithmically prior to data plotting and data analysis. Paired or unpaired *t*-tests were used for two group comparisons. If samples had unequal variances (determined by an *F* test), an unpaired *t*-test with the Welch correction was used. For multiple-group

comparisons, unequal-variance versions of ANOVA (one-way or two-way ANOVA) were performed, followed by a Tukey's multiple comparison test with the exception that for repeated measurements two-way ANOVA was followed by a Sidak's multiple comparison test. Survival curves were estimated from the Kaplan–Meier procedure with the log-rank (Mantel–Cox) test to compare survival among groups. All tests used were two-tailed. The significance threshold for all tests was set at the 0.05 level.

**Expanded View** for this article is available online.

## Acknowledgements

This project was funded by the Foundation for Experimental Biomedicine Zurich, Switzerland, to AB, by the German Research Foundation DFG VO 1602/4-1 to AB and DFG KA 1797/7-1 to ZK. NA was supported by a release from teaching grant from the Deutsches Zentrum für Herz-Kreislaufforschung. CCG was supported by a MD scholarship provided by the Berlin Institute of Health. We acknowledge Nadine Albrecht-Köpke, Sandra Bundschuh, and Martina Sauter for excellent technical assistance. LMP7$^{-/-}$ mice on a C57BL/6 background were originally provided by Ulrich Steinhoff. We acknowledge Armin Rehm, Arturo Zychlinski, and Borko Amulic for their support on neutrophil function. Florian Leuschner and Kaweeh Molawi are kindly acknowledged for sharing their expertise on innate immune cells in heart tissue.

## Author contributions

Conceptualization: AB, CCG, NA; formal analysis: NA, CCG, KK, SP; funding acquisition: AB, ZK, NA, CCG; investigation: NA, CCG, KK, KV, MK, AH, SP; methodology: MK, AH, NA, CCG, SP; project administration: AB; resources: AB, ZK, AH; supervision: AB, ZK; visualization: NA, CCG, MK, KK, SP; writing and/or original draft: AB, NA, CCG.

## Conflict of interest

The authors declare that they have no conflict of interest.

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
