## [Review Process File · EMBO Molecular Medicine]

The immunoproteasome-specific inhibitor ONX 0914 reverses susceptibility to acute viral myocarditis

Nadine Althof, Carl Christoph Goetzke, Meike Kesphohl, Karolin Voss, Arnd Heuser, Sandra Pinkert, Ziya Kaya, Karin Klingel, and Antje Beling

Review timeline:

Submission date:	29 May 2017
Editorial Decision:	03 July 2017
Revision received:	01 November 2017
Editorial Decision:	23 November 2017
Revision received:	28 November 2017
Accepted:	01 December 2017

Editors: Roberto Buccione & Céline Carret

Transaction Report:

1st Editorial Decision

03 July 2017

Thank you for the submission of your manuscript to EMBO Molecular Medicine and apologies for the delay in providing you with a decision. We experienced difficulties in securing three willing and appropriate reviewers and furthermore, their evaluations were delivered with some delay.

We have now heard back from the three expert Reviewers whom we asked to evaluate your manuscript.

As you will see, the reviewers agree that manuscript presents potentially interesting findings. However, reviewers 1 and 2 also point to serious and partially overlapping concerns that can be summarised as follows: 1) lack of analysis/insight into the molecular mechanisms of the protection conferred by ONX 0914 treatment; 2) lack of demonstration of the purported anti-viral function of ONX 0914; 3) lack of demonstration that immune-mediated injury is reduced after ONX 0914 treatment and 4) the contradiction with your earlier findings is not addressed/explained.

After our reviewer cross-commenting exercise, there was agreement that while the above concerns (in addition to the other items) would need to be resolved, you may address the molecular mechanism either in in vivo infection experiments with conditional gene-deficient mice, or in vitro (using cell cultures).

In conclusion, while publication of the paper cannot be considered at this stage, given the potential interest of your findings, we have decided to give you the opportunity to address the criticisms.

We are thus prepared to consider a substantially revised submission, with the understanding that the Reviewers' concerns must be addressed with additional experimental data where appropriate as indicated above, and that acceptance of the manuscript will entail a second round of review.

I understand that if you do not have the required data available at least in part, to address the above, this might entail a significant amount of time, additional work and experimentation and might be technically challenging, I would therefore understand if you chose to rather seek publication

elsewhere at this stage. Should you do decide to do so, and we hope not, we would welcome a message to this effect.

***** Reviewer's comments *****

Referee #1 (Remarks):

Althof and colleagues wrote a very interesting manuscript on the role of the proteasome inhibitor ONX 0914 in a mouse model of viral myocarditis. After treating susceptible A/J mice with ONX 0914, in the early viral phase the authors observed reduced mortality, lower inflammatory cells and inflammatory cytokines/chemokines, and improved cardiac function in the heart, but higher T cells, neutrophils and monocytes/macrophages in the spleen. The potential of these findings is big, and the authors tried to explain their observations in an almost proper way. However, I have some concerns, questions, and comments.

1 - The major weak point of this manuscript is that the authors did not figure out the mechanism underlying their findings. The authors did not describe the protective molecular mechanisms conferred by ONX 0914 treatment.

2 - It is sometimes difficult to understand why the authors jump from experiments with CVB3 infection to experiments with ONX 0914 treatment only. It would be easier for the reader to see first what is the effect of ONX 0914 on mice without viral challenge, and then see all the experiments performed with CVB3 to determine the role of ONX 0914 after viral challenge.

3 - The authors have never demonstrated the anti-viral function of ONX 0914 in this manuscript. They just mention in few sentences on pages 5 and 6 that they found almost irrelevant levels of IFN-beta in both ONX 0914- and vehicle-treated mice. Other potential anti-viral cytokines, such as IFN-alpha or IFN-gamma, have not been mentioned, while IL-6, which has also been described to have anti-viral effects in CVB3-induced myocarditis, has been measured at the RNA level in heart tissue and in serum, but in both cases it was lower than in control mice. The authors mention in the abstract, line 12, that "... in turn initiated a strong and effective antiviral adaptive response.", but this has been just partially shown and demonstrated in this manuscript. Just showing higher production of anti-CVB3 antibodies at day 8 does not explain the reduced mortality of ONX 0914-treated mice between day 5 and day 6. The authors should show the same antibody production at day 6, if this is the anti-viral adaptive response they are talking about.

4 - What is the real protective mechanism induced by ONX 0914? It would be really interesting to understand which cells or molecules would induce protection and reduce mortality in the early viral myocarditis phase. I frankly doubt that highly activated neutrophils and macrophages, which have higher phagocytosis capacity, are the only reason for reduced mortality and lower viral titres in the heart in the first days after CVB3 infection.

5 - In figure 4 the authors challenged C57BL/6 mice with CVB3. Neither the mortality rate has been shown or mentioned, nor if any mortality has been observed. To corroborate their findings, the authors should show the anti-CVB3 antibody titres in both groups, although they simply mention that they were able to detect CVB3 neutralizing antibodies in both groups of mice.

6 - The authors did not explain why, after CVB3 infection, lymphocytes in the heart were reduced in ONX 0914-treated mice, while in spleen and blood of the same mice lymphocytes were increased.

7 - The authors never explain which is the real anti-viral factor that reduces viral titres in the heart two days after CVB3 infection in the ONX 0914 group. In addition, virus titres in the spleen, which has been widely described in terms of number of leukocytes, have just been shortly mentioned in the discussion section, on page 12, line 4. The authors should show these data to better understand if the higher numbers and increased activation of neutrophils in the spleen are able to directly reduce viral load. Otherwise the effect of neutrophils and monocytes/macrophages described in the manuscript is just speculative.

8 - The authors state that the detrimental immune-mediated injury was reduced after ONX 0914 treatment. To demonstrate it, the authors should show if apoptosis or necrosis in the heart was reduced, or if cytotoxic signals were attenuated in protected mice.

9 - In the discussion section, the authors' motivations about the anti-viral effects of ONX 0914 are speculative. The authors should also consider that other cells, such as for example natural killer cells and cytotoxic CD8 T cells, by producing perforin and granzymes, have an anti-viral function in viral myocarditis. In addition, plasmacytoid dendritic cells, which produce high amounts of IFN-alpha, are also able to limit viral proliferation in the same mouse model.

10 - In the discussion section, on page 13, lines 5 and 6, the authors state that the "...the mononuclear phagocytes represented the predominant infiltrating cell type during acute stage of

heart tissue infection". Although the amount of Mac3 and CD11b cells has been compared between captisol- and ONX 0914-treated mice challenged with the CVB3 virus, no further extended cell population analysis of heart infiltrates has been performed. Therefore, the sentence is not appropriate. Please provide a larger analysis of infiltrating cells or just add a reference to support your statement.

11 - In the discussion section, on page 13, lines 13-16, it is true that high concentrations of TNF-alpha, IL-6, and IL-1beta are detrimental for the heart tissue, but it is also true that IL-6 contributes to limit viral expansion and therefore it exerts a beneficial anti-viral effect.

12 - Minor comment - In addition to analysis of heart function at day 8, it would be also interesting to determine heart function at day 4 or 5, just before the majority of vehicle-treated mice succumbed.

13 - Minor comment - On Figure 4A, it would be better to have the same scale with the same concentrations on the Y-axis. It would be easier to understand that the differences in T cells are significant at day 0, while at day 8 the differences are not significant.

Referee #2 (Comments on Novelty/Model System):

The used CVB3 model system seems to lead to contradictory results, one time arguing in favor and one time against a role for immunoproteasomes in amelioration of CVB3-caused heart muscle injury.

Referee #2 (Remarks):

This manuscript examines whether treatment with an LMP7-specific proteasome inhibitor, ONX 0914, may limit heart pathology in CVB3-infected mice. The authors show that ONX treatment enhances early release of neutrophils and monocytes into the blood / spleen and enhances phagocytic function of these cells. Both chemokine and cytokine release, and importantly, infiltration of myeloid cells into heart tissue are reduced. Viral titers are less and cardiac function is improved, leading to the conclusion that systemic immunoproteasome (LMP7) inhibition by ONX0914 may represent an attractive means for treatment of severe virus-mediated inflammation of the heart.

These findings directly contradict a previous publication by the same authors, demonstrating that 'impairment of immunoproteasome function by LMP7 subunit deficiency results in severe enterovirus myocarditis' (ref 23). In this publication, it is shown that immunoproteasome deficiency is associated with severe acute heart muscle injury reflected by large foci of inflammatory lesions and severe myocardial damage. This injury is attributed to proteotoxic stress, resulting from impaired removal of poly-ubiquitinated protein aggregates in the absence of immunoproteasomes. In the mouse model used in this publication, proteasomes possessed six functional catalytic sites (LMP7 replaced by beta 5), while in the present manuscript, two of the catalytic sites of immunoproteasomes are blocked by ONX 0914.

This discrepancy between data in the present compared to the previous manuscript should be carefully explained.

In addition, any mechanistic insight into how described differences in innate immune cell presence and function, between ONX-treated and untreated infected mice, influence CVB3-caused pathology is lacking. If claiming an explanatory role for neutrophils and monocytes in the blood in early control of CVB3 infection in ONX-treated mice, additional experiments, for example with conditional KO mice, are needed to demonstrate the suggested role of these cells in amelioration of CVB3-induced disease.

Minor comments:

Fig 4B, B6 mice: please show weights of treated and untreated mouse groups over the full course of CVB3 challenge.

Neutrophil / monocyte release from the BM is caused by chemokines. Please show whether serum

levels of responsible chemokines are enhanced by ONX treatment.

Ref to Fig 3A, B at p. 8 is misplaced

Referee #3 (Comments on Novelty/Model System):

Overall, experiments were executed very well and all the experiments were appropriate to support the claims by authors: highly relevant model and assays.

Referee #3 (Remarks):

The role of immunoproteasome in viral myocarditis is highly interesting and authors used a LMP7 inhibitor to test it. All the experiments were performed well with appropriate controls and the results obtained were very supportive of important role of immunoproteasome in the disease model. An additional control, in addition to captisol, that might further convince the role of LMP7 in viral myocarditis is the use of inactive ONX 0914 (an epimer with different stereochemistry at the epoxide ring). While the use of epimer may further convince the findings described in this manuscript, the work overall is well done and worth publication without revision.

1st Revision - authors' response

01 November 2017

Overall achievements of the revision

(A) Concerns on contradictory findings of CVB3-mediated acute myocarditis in B6-LMP7^{-/-} mice.

Initially, we would like to share some general aspects on the viral myocarditis model that we find important concerning our choice of the A/J strain in this manuscript. The disease outcome of viral myocarditis in humans may range from subclinical disease to severe illness with cardiac death (Cooper, 2009; Sagar et al, 2012). The genetic background and associated with this the immunological equipment of the host are thought to be predisposing factors for a devastating outcome. Such adverse immune response activation – also referred to immunopathology – can be mimicked in mouse models with immune-anchored high susceptibility to viral myocarditis (Klingel et al, 1992). Since we were specifically interested in elucidating targets to counteract this devastating immunopathology, we conducted our experiments in A/J mice with a high hereditary susceptibility to viral myocarditis. In clear contrast to the A/J strain, C57BL/6 mice (B6) mice can be considered as model organisms for patients with mild acute disease and are resistant in terms of development of clinically relevant viral cardiomyopathy upon infection with a cardiotropic CVB3 (Nancy) strain (Corsten et al, 2012; Esfandiarei & McManus, 2008; Jakel et al, 2009; Klingel et al, 1992; Szalay et al, 2006). This diverse disease pattern is evident upon inoculation of 10⁵ PFU CVB3 Nancy in B6 mice and 10⁴ PFU (1-log-fold less virus) in A/J mice.

In 2011, we reported on exacerbated inflammation of the heart muscle in response to genetic deletion of the LMP7 subunit in mice of a C57BL/6 background – a as said resistant deemed genetic background regarding severity of virus-induced heart tissue damage (Opitz et al, 2011). Although disease exacerbation was clearly evident in B6-LMP7^{-/-} mice, these mice in clear contrast to A/J mice still ranked as resistant in terms of clinical relevant viral cardiomyopathy. This is reflected by maintained survival, cardiac function and complete disease resolution after long-term observation (Opitz et al, 2011). Reviewer 2 raised the need to discuss contradictory results between LMP7 deletion and ONX 0914 treatment in the present study. This prompted us to investigate the effects induced by ONX 0914 also in B6 wild-type mice where, other than in genetic LMP7 deficiency, only the LMP7 subunit was targeted. The as robust described phenotype in B6-LMP7^{-/-} mice in terms of mortality as well as the increased infiltration into infected heart tissue could be confirmed

under ONX 0914 influence in WT B6 mice (Fig 1). In terms of overall pathology, our findings in ONX 0914-treated B6 mice corroborated our previous data on disease exacerbation, yet in comparison to A/J mice still mild disease course in B6-LMP7^{-/-} mice (Opitz et al, 2011). Especially type I interferon (T1IFN) signaling into cardiomyocytes contributes substantially to mild disease in B6 mice (Althof et al, 2014; Rahnefeld et al, 2014). Consistent with previous work (Ichikawa et al, 2012; Muchamuel et al, 2009) ONX 0914 suppressed T1IFN responses in B6 mice during infection (Fig 1F). In line with this, LMP7 inhibition by ONX 0914 in wt B6 mice resulted in increased viral load in the heart (Fig 1E). In addition to these novel findings, we also discussed proposed mechanisms that contribute to exacerbated inflammation in B6-LMP7^{-/-} mice (Ebstein et al, 2013; Opitz et al, 2011; Paeschke et al, 2016; Seifert et al, 2010) in comparison to ONX 0914 treated B6 and A/J mice. As already reported in the original manuscript, activation of systemic T1IFN responses is inherently reduced in A/J mice with only mild additive action of ONX 0914 (Fig EV 3). In A/J mice, viral infection triggers detrimental immune responses with overwhelming cytokine/chemokine production being reminiscent of a sepsis-like course. Devastating immune responses are central for disease severity in this host. ONX 0914 drastically diminishes this immunopathology (Fig 2, 7, 8).

These different aspects of immunoproteasome inhibition in A/J and B6 mice are carefully addressed and discussed in the revised manuscript. Also, we provide additional data demonstrating poly-ubiquitin conjugates in inflamed mouse hearts from A/J mice. Based on these data, we propose a minor impact of immunoproteasome proteolysis for maintaining proteostasis in this highly susceptible host.

(B) Concerns on the purported anti-viral function of ONX 0914

Our initial interpretation of a direct anti-viral function under ONX 0914 influence was diffuse. Prompted by this concern, we followed viral titers in different organs and at different points in time (Fig 3). Based on these data, we have sharpened our interpretation/conclusion on the relationship between ONX 0914 treatment and viral replication, dissemination, and control by the immune system. Also, we have conducted additional experiments addressing the function of neutrophils for control of CVB3.

The key findings are:

- No experimental evidence for a direct antiviral function of ONX 0914.
- Neutrophils are not substantially involved in CVB3 control.
- Virus spreading to heart and spleen is delayed, which is attributed to reduced immune-mediated pancreatic tissue damage
- Viral burden in heart and pancreas at the respective organ-specific peak of viral cytotoxicity not influenced by ONX 0914
- Viral burden during the replicative phase in pancreas and spleen is not affected.

Taken together, ONX 0914-induced protection from virus-induced pathology in A/J mice appears not to be attributed to a major antiviral effect and its positive sequels.

Immunoproteasome inhibition had no direct effect on the replication machinery of the virus *in vitro* (Fig EV1B). We initially demonstrated that ONX 0914 treatment delayed viral dissemination into heart and spleen early after infection. To address the biological impact of this finding regarding to the drastic improvement of overall pathology, we followed viral titers during disease course in these organs and in serum. Since CVB3 initially targets the pancreas after intraperitoneal inoculation and initially replicates in this organ, virus titers were determined in pancreas as well. We found that viral

replication in the pancreas was unaffected, but viral dissemination from the pancreas to serum and other organs like spleen and heart was delayed yielding no substantial differences between infectious viral titers at day 3 and 4 (Fig 3). Since ONX 0914 has no substantial effect on primary virus replication in the pancreas (peak day 2) and on infectious virus particle formation in heart tissue (peak day 4-6) (Fig 3), we conclude that ONX 0914 does not directly influence CVB3 replication in pancreas and heart.

To investigate whether an increased abundance of activated neutrophils in spleen and blood might contribute to the observed delayed virus spreading in spleen or heart, we investigated a defined function of neutrophils that also operative for viral pathogens. Activation of neutrophils induces the formation of neutrophil extracellular traps (NETs). Other than reported for some enveloped viruses, NETosis did not directly impact the non-enveloped CVB3 (Fig 5G). To get more insights into neutrophil function in A/J mice, next we depleted neutrophils with Ly6G antibodies prior to CVB3 infection. Neutrophil depletion was well tolerated during CVB3 infection yielding no differences of viral titers 2, 3 and 8 days after infection (Fig 5H, EV5). Overall, we conclude that the ONX 0914-induced increased abundance of activated neutrophils in spleen and blood does not directly contribute to the delay of virus dissemination found in heart and spleen.

Although delayed virus dissemination in heart and spleen most likely only contributes to a minor degree if at all to substantial diminished pathology under ONX 0914 influence, we questioned which aspects might be involved in this delay. Since reduced viral loads in serum, heart and spleen paralleled temporally with the peak of peak virus replication in the pancreas (Fig 3), we hypothesized that virus-induced immunopathology was also diminished in the pancreas as found for heart tissue by ONX 0914 treatment. Indeed, we found reduced serum lipase activity in ONX 0914-treated mice, which mirrors reduced death of exocrine pancreatic cells at this point in time (Fig EV1). Thereby, this diminished tissue destruction might lead to a delayed release of infectious viral particles, which is in line with the temporally reduced viral burden in serum, heart and spleen at day 2.

We agree with reviewer 1 that – while the adaptive immune response was improved as reflected by increased formation of CVB3-neutralizing antibodies 8 days after infection in ONX 0914-treated A/J mice (Fig 2G) – this effect does not substantially contribute to improved survival and reversal of the distributive shock condition found in vehicle-treated mice. The protective effector function of neutralizing antibodies is critical at later stages.

(C) Concerns on lack of demonstration that immune-mediated injury is reduced after ONX 0914 treatment in A/J mice

From our perspective, we provide several lines of evidence that clearly demonstrate a substantial reduction of immune-mediated injury (immunopathology) by ONX 0914 in infected A/J mice.

The major findings are:

- Reduction of inflammatory cell burden in heart tissue:

The inflammatory cell burden during virus-mediated myocarditis determines disease outcome (Kindermann et al, 2008). Monocytes and macrophages are centrally involved in mediating tissue damage and fibrotic scarring (Hirasawa et al, 1996; Jaquenod De Giusti et al, 2015). Infiltration of monocytes/macrophages in CVB3-infected A/J mice under the influence of ONX 0914 was quantified by (i) immunohistochemistry (quantification of Mac-3 signals) and (ii) flow cytometry of myeloid cells (Fig 2C-E). The latter method uses intact organs, and therefore it is advantageous since the issue of an existing sampling error can be neglected. Also, (iii) local

cytokine/chemokine expression reflects immunopathology since most of the respective cytokines/chemokines are produced by infiltrating monocytes/macrophages (Fig 7C).

- Protection from overwhelming cytokine-storm in A/J mice → suppression of pro-inflammatory cytokine/chemokine production by ONX 0914. In addition to serum (Fig 7B) and heart tissue data (Fig 7C), we have also determined inflammatory cytokine/chemokine production in spleen 3 days p.i. since ONX 0914 substantially influenced the abundance of immune cells during infection (Fig 7A).
 - (i) Spleen (d3 p.i.): TNF- α , IL-6, IL-1 β , chemokines (Fig 7A)
 - (ii) Systemic response (serum, ELISA/Multiplex assay; d3 p.i.): TNF- α , IL-6, IL-1 β , chemokines (Fig 7B)
 - (iii) Heart (d8 p.i.): TNF- α , IL-6, IL-1 β , chemokines (Fig 7C), IFN- γ (Fig 2H)
- Cytotoxic T-cell activation / immunopathology in the heart:
Parameters of T cell activation were determined during myocarditis. IFN- γ , granzyme A and perforin 1 were all reduced in heart tissue of ONX 0914-treated mice (Fig 2H).
- The mean fluorescence intensity of Fixable Viability Dye⁺/CD45⁻ cells in heart tissue was reduced by ONX 0914, which is another indicator of reduced cell death in the inflamed mouse heart (Fig 2G).
- In vehicle-treated mice, we observed a sepsis-like course of the disease mirrored by a significant hemodynamic compromise. The hemodynamic condition was reminiscent of a distributive shock condition found in sepsis-like condition. Distributive shock is usually attributed to an overwhelming cytokine response. Consistently, clinical studies reported on the adverse effects of pro-inflammatory cytokines in humans suffering septic shock revealing both a cardio-depressive effect of TNF- α and IL-1 (Cain et al, 1999; Kumar et al, 1996). ONX 0914-mediated suppression of cytokine production and attenuation of immunopathology are the driving forces for improvement of cardiac function in infected A/J mice. These functional hemodynamic parameters are a direct measure of reversed immunopathology upon ONX 0914 treatment (Table 1).
- Likewise, improved survival and reduced body weight were detected under ONX 0914 influence (Fig 2A/B). Since the viral burden was affected only to a minor extent and temporally, the overall improvement of infection pathology (body weight, survival, cardiac performance) directly mirrors ONX 0914-mediated effects on immune response activation.

(D) Concerns on molecular insights into the protective influence exerted by ONX 0914 in CVB3-infected A/J mice.

In the revised manuscript, we emphasize that a suppression of detrimental adverse immune response activation is central for the overall beneficial effects exerted by ONX 0914. The main aspect is the global (serum, spleen) as well as heart-tissue specific suppression of pro-inflammatory cytokine/chemokine production (Fig 7) and thereby mediated effects (Table 1, Fig 2). ONX 0914 independently of virus inoculation mobilized inflammatory monocytotic cells from bone marrow sources and increased their abundance in spleen. Nevertheless, cytokine/chemokine effector function in addition to many other aspects is needed to actually induce migration and infiltration of these cells into heart tissue.

Since monocytes/macrophages are centrally involved in mediating tissue damage in the heart and represent important sources of cytokines during infection (Corsten et al, 2012; Jaquenod De Giusti et al, 2015), we focused on ONX 0914-mediated molecular aspects of suppressed

cytokine/chemokine production in cell culture studies. Thereby, experimentally bone marrow-derived macrophages (BMM) were challenged with artificial single-stranded RNA – a Toll-Like Receptor 7 (TLR7) agonist – mirroring natural conditions under which intracellular coxsackieviral RNA genomes serve as pathogen-associated molecular patterns (PAMPs) and induce strong cytokine expression in monocytes/macrophages provided by the bone marrow in response to virus encounter. TLR7-activated BMM responded with increased TNF- α , IL-1 β , IL-6, MCP-1, IP-10 and MIP-2 α production (Fig 8). TLR7-engagement allows binding between TLR7 and MyD88, which then induces MAPkinase signaling (Blasius & Beutler, 2010). We investigated whether MAPkinases might be involved and initially used specific inhibitors of the three major MAPkinases ERK1/2, p38 and JNK to elucidate a putative contribution of the respective kinases to TLR7-induced cytokine/chemokine production. Fig 8A illustrates a role of MAPkinase activation particularly for IL-6 and IL-1 β production. Moreover, we found that SP600125, which interferes with JNK/p38 phosphorylation, also reduced TNF- α and MIP-2 α expression (Fig 8A). Next, we questioned whether as observed for spleen and heart tissue during infection, ONX 0914 treatment influences cytokine/chemokine production also in TLR7-activated BMM. We found a profoundly suppressed cytokine/chemokine mRNA induction in cells that had been treated with ONX 0914 (Fig 8B). Since MAPkinases were indeed involved in TLR7-dependent cytokine induction and ONX 0914 negatively influenced TLR7-induced responses, we pursued the hypothesis that ONX 0914 affects MAPkinase activation upon TLR7 engagement. In fact, under the influence of ONX 0914 the abundance of the respective phosphorylated kinases p-p38, p-ERK1/2 and p-JNK was significantly reduced in TLR7 agonist-treated cells (Fig 8C/D).

From these experiments, we conclude that PAMP-induced activation (ssRNA) of respective Pattern Recognition Receptors (PRRs) (TLR7) and thereby affected MAPkinase signaling events in BMM are influenced by ONX 0914 inhibition. Reduced activation of MAPkinase signaling paralleled in reduced cytokine/chemokine production, which in turn was found to be central for ONX 0914-mediated protective effects. We demonstrated and discussed these novel findings in the revised manuscript appropriately.

Point-by-point response

Reviewer 1:

Althof and colleagues wrote a very interesting manuscript on the role of the proteasome inhibitor ONX 0914 in a mouse model of viral myocarditis. After treating susceptible A/J mice with ONX 0914, in the early viral phase the authors observed reduced mortality, lower inflammatory cells and inflammatory cytokines/chemokines, and improved cardiac function in the heart, but higher T cells, neutrophils and monocytes/macrophages in the spleen. The potential of these findings is big, and the authors tried to explain their observations in an almost proper way. However, I have some concerns, questions, and comments.

Concern 1: The major weak point of this manuscript is that the authors did not figure out the mechanism underlying their findings. The authors did not describe the protective molecular mechanisms conferred by ONX 0914 treatment.

Response: In the revised manuscript, we have specifically highlighted the substantial immunomodulatory influence of ONX 0914. The reversal of virus-induced immunopathology is attributed to an overall maintenance of immune homeostasis that contribute to low-grade immune-mediated tissue injury and efficient antiviral immune responses found in compound-treated mice.

More detailed information is provided in the first section on overall achievements:

(D)Concerns on molecular insights into the protective influence exerted by ONX 0914

in CVB3-infected A/J mice and
 (C) Concerns on lack of demonstration that immune-mediated injury is reduced after
 ONX 0914 treatment in A/J mice.

Concern 2: It is sometimes difficult to understand why the authors jump from experiments with CVB3 infection to experiments with ONX 0914 treatment only. It would be easier for the reader to see first what is the effect of ONX 0914 on mice without viral challenge, and then see all the experiments performed with CVB3 to determine the role of ONX 0914 after viral challenge.

Response: We have rearranged the manuscript according to the recommendation of the reviewer (Fig 4, 5, 6).

Concern 3: The authors have never demonstrated the anti-viral function of ONX 0914 in this manuscript. They just mention in few sentences on pages 5 and 6 that they found almost irrelevant levels of IFN-beta in both ONX 0914- and vehicle-treated mice. Other potential anti-viral cytokines, such as IFN-alpha or IFN-gamma, have not been mentioned, while IL-6, which has also been described to have anti-viral effects in CVB3-induced myocarditis, has been measured at the RNA level in heart tissue and in serum, but in both cases it was lower than in control mice. The authors mention in the abstract, line 12, that "... in turn initiated a strong and effective antiviral adaptive response.", but this has been just partially shown and demonstrated in this manuscript. Just showing higher production of anti-CVB3 antibodies at day 8 does not explain the reduced mortality of ONX 0914-treated mice between day 5 and day 6. The authors should show the same antibody production at day 6, if this is the anti-viral adaptive response they are talking about.

Response: We are in agreement with the reviewer that aspects of a direct anti-viral property of ONX 0914 treatment were not well addressed in the original manuscript. In the revised manuscript, we demonstrate several additional experiments on the viral burden in vehicle- and ONX 0914-treated mice. Detailed information on this issue is presented in the first section on overall achievements → (B) Concerns on the purported anti-viral function of ONX 0914.

In addition, we demonstrate data on T1IFN induction in A/J mice (Fig EV3). IFN- γ expression in heart tissue is demonstrated in Fig 2F. Systemic IFN- γ production was below the detection limit (data not shown). IL-6 production was found to be substantially reduced under ONX 0914 influence in serum, spleen, heart tissue and in TLR7-activated bone marrow derived macrophages (Fig 7, 8). For further discussion on the role of IL-6, please refer to concern 11.

As stated above, we have sharpened our interpretation/conclusion on the relationship between ONX 0914 treatment and viral replication, dissemination, and elimination of the pathogen by the immune system. We agree with the reviewer that – while the adaptive immune response was considerably improved as reflected by increased formation of CVB3-neutralizing antibodies 8 days after infection in ONX 0914-treated A/J mice (Fig 2G) – this effect does not substantially contribute to improved survival and reversal of the distributive shock condition found in vehicle-treated mice. If biologically significant for diminished pathology, the protective effector function of neutralizing antibodies would need to commence in different virus load at points in time where viral pathology peaks. However, viral burden was not influenced by ONX 0914 at stages where survival, body loss (beginning around day 5-6) as well as cardiac performance (day 8) were found to be beneficially affected by ONX 0914. These aspects are discussed appropriately.

Concern 4: What is the real protective mechanism induced by ONX 0914? It would be really interesting to understand which cells or molecules would induce protection and reduce mortality in the early viral myocarditis phase. I frankly doubt that highly activated neutrophils and macrophages,

which have higher phagocytosis capacity, are the only reason for reduced mortality and lower viral titres in the heart in the first days after CVB3 infection.

Response: We are in agreement with the reviewer that activated neutrophils and macrophages do not substantially explain the highly beneficial phenotype under ONX 0914 influence. Also, investigation of the biological impact of neutrophils during the revision process revealed that neutrophils, which were substantially influenced regarding abundance and function by ONX 0914, most likely do not majorly attribute to diminished immunopathology found in compound-treated A/J mice.

More detailed information is provided in the first section (overall achievements) on which molecules and which cells are presumably involved in protection from overall pathology.

(C) Concerns on lack of demonstration that immune-mediated injury is reduced after ONX 0914 treatment in A/J mice.

(D) Concerns on molecular insights into the protective influence exerted by ONX 0914 in CVB3-infected A/J mice

Reversal of susceptibility to severe disease can be attributed to:

- Suppression of systemic and cardiac pro-inflammatory cytokine/chemokine production
- Reduced tissue damage in heart and presumably also in pancreas due to maintained immune homeostasis in response to virus encounter
- Improved cardiac output as a result of reduced distributive shock condition
- Reduced body weight loss might involve reduced pancreatic tissue damage.

In the revised manuscript, we are discussing the beneficial role of ONX 0914 on the overall dominating adverse immune response activation, which has local effects in the hearts (myocardial injury and function) and also influences the overall pathology (body weight, survival, distributive shock).

Concern 5: In figure 4 the authors challenged C57BL/6 mice with CVB3. Neither the mortality rate has been shown or mentioned, nor if any mortality has been observed. To corroborate their findings, the authors should show the anti-CVB3 antibody titres in both groups, although they simply mention that they were able to detect CVB3 neutralizing antibodies in both groups of mice.

Response: The requested data are shown in the revised manuscript (Fig 1A – survival, Fig 1F – antibody titers).

Concern 6: The authors did not explain why, after CVB3 infection, lymphocytes in the heart were reduced in ONX 0914-treated mice, while in spleen and blood of the same mice lymphocytes were increased.

Response: We demonstrated that ONX 0914 treatment reduced peripheral lymphocytes, which is in agreement with published data (Hensley et al, 2010) (Fig 4A/B). Upon infection with CVB3, vehicle-treated mice develop a lymphopenia in blood and spleen until 4 days p.i., whereas ONX 0914 treatment prevented this lymphopenia (Fig 4C/D). These differences are not evident after 8 days (Fig 4E). After 8 days, the overall abundance of lymphocytes was not significantly affected by ONX 0914 in infected mice. We apologize for this confusion, which might originate from the initial way our data were presented. Since we have rearranged the manuscript according to your suggestions, this aspect is hopefully much clearer now.

Concern 7: The authors never explain which is the real anti-viral factor that reduces viral titres in the heart two days after CVB3 infection in the ONX 0914 group. In addition, virus titres in the spleen, which has been widely described in terms of number of leukocytes, have just been shortly mentioned in the discussion section, on page 12, line 4. The authors should show these data to better

understand if the higher numbers and increased activation of neutrophils in the spleen are able to directly reduce viral load. Otherwise the effect of neutrophils and monocytes/macrophages described in the manuscript is just speculative.

Response: Detailed information particularly on the biological impact of ONX 0914-induced neutrophilia is presented in the first section on overall achievements → (B) Concerns on the purported anti-viral function of ONX 0914 (Fig 5G/H and Fig EV5). Virus titer in different organs and in serum are illustrated in Fig 3.

Concern 8: The authors state that the detrimental immune-mediated injury was reduced after ONX 0914 treatment. To demonstrate it, the authors should show if apoptosis or necrosis in the heart was reduced, or if cytotoxic signals were attenuated in protected mice.

Response: Apoptosis did not substantially contribute to tissue injury (Fig EV4B). The mean fluorescence intensity of Fixable Viability Dye⁺/CD45⁻ cells in heart tissue was reduced by ONX 0914, which is an indicator of reduced cell death in the inflamed mouse heart (Fig 2G).

Concern 9: In the discussion section, the authors' motivations about the anti-viral effects of ONX 0914 are speculative. The authors should also consider that other cells, such as for example natural killer cells and cytotoxic CD8 T cells, by producing perforin and granzymes, have an anti-viral function in viral myocarditis. In addition, plasmacytoid dendritic cells, which produce high amounts of IFN- α , are also able to limit viral proliferation in the same mouse model.

Response: Following requests and recommendations of the reviewers, we could sharpen our conclusions in the discussion section. Please also refer to concern 7. Parameters of T cell activity were determined during myocarditis. IFN- γ , granzyme A and perforin 1 were all reduced in heart tissue of ONX 0914 treated mice (Fig 2H) and reflect the reduced activation status T cells are in within this group during infection. NK cell abundance is considerably low in A/J mice during infection and only insignificantly influenced by ONX 0914 (Figure shown on next page).

Concern 10: In the discussion section, on page 13, lines 5 and 6, the authors state that the "...the mononuclear phagocytes represented the predominant infiltrating cell type during acute stage of heart tissue infection". Although the amount of Mac3 and CD11b cells has been compared between captisol- and ONX 0914-treated mice challenged with the CVB3 virus, no further extended cell population analysis of heart infiltrates has been performed. Therefore, the sentence is not appropriate. Please provide a larger analysis of infiltrating cells or just add a reference to support your statement.

Response: The requested data demonstrating highest abundance of mononuclear phagocytes in comparison to other immune cells at day 8 after infection is shown below. We have included a suitable reference in the revised manuscript: (Klingel et al, 1992; Opitz et al, 2011).

Concern 11: In the discussion section, on page 13, lines 13-16, it is true that high concentrations of TNF-alpha, IL-6, and IL-1beta are detrimental for the heart tissue, but it is also true that IL-6 contributes to limit viral expansion and therefore it exerts a beneficial anti-viral effect.

Response: Undisputedly, IL-6 production might exert beneficial effects on pathogen clearance. Nevertheless, as shown for other pro-inflammatory cytokines and stated in the revised manuscript, IL-6 concentration as well as the timing of its release determines how this cytokine affects the outcome of virus-induced myocarditis (Corsten et al, 2012). Two papers actually of the same group reported on IL-6 function during viral myocarditis and revealed somewhat controversial findings: In the first report, the authors demonstrated that exogenous IL-6 administration, given at the time of virus inoculation, has a protective effect on myocardium and improves survival rates (Kanda et al, 1996). In their second paper, the authors used IL-6 transgenic mice (IL-6TG) and reported severe myocardial injury in IL-6TG, which was consistent with an increased viral load in the heart (Tanaka et al, 2001). This study actually reflects the situation that we found both in ONX 0914-treated A/J mice, where we found reduced IL-6 levels and improved disease. Douglas Mann described the impact of IL-6 and viral myocarditis to the point: the Yin-Yang of cardiac innate immune responses (DOI: 10.1006/jmcc.2001.1432).

Minor concern 12: In addition to analysis of heart function at day 8, it would be also interesting to determine heart function at day 4 or 5, just before the majority of vehicle-treated mice succumbed.

Response: We are in agreement with the reviewer that it might be interesting to obtain further insights into the course of cardiac function at earlier points in time as well. Consistent with the detection of the systemic cytokine storm at these points in time, we would expect to confirm our day 8 finding of the distributive shock condition, vehicle-treated A/J mice suffer from, also at earlier states. Saying that, we need to emphasize once more that the diastolic underfilling found in vehicle-treated A/J mice majorly contributes to reduced cardiac output and originates from a distributive shock condition mediated by pro-inflammatory cytokines either directly or by indirect mechanisms. Since myocardial injury in viral myocarditis involves infiltration and viral cytotoxicity, in terms of cardiac performance we expect most impressive alterations at the state of most severe myocarditis (8 days p.i.) as well. Given that we expect to face significant issues to justify the suggested experiments (at least 20 mice would be needed) at local authorities.

Nevertheless, mortality is evident prior to day 8 p.i., where we assessed cardiac performance in A/J mice. In the revised manuscript, we strengthened this aspect in the discussion section and highlighted systemic pathology, A/J mice suffer from upon virus inoculation and which was reversed by ONX 0914 treatment. To the end, we hope that the reviewer shares our view that most likely no substantial novel information would be obtained from assessment of cardiac function at earlier states.

Minor concern 13: On Figure 4A, it would be better to have the same scale with the same concentrations on the Y-axis. It would be easier to understand that the differences in T cells are significant at day 0, while at day 8 the differences are not significant.

Response: The comment was addressed accordingly.

Reviewer 2:

The used CVB3 model system seems to lead to contradictory results, one time arguing in favor and one time against a role for immunoproteasomes in amelioration of CVB3-caused heart muscle injury. This manuscript examines whether treatment with an LMP7-specific proteasome inhibitor, ONX 0914, may limit heart pathology in CVB3-infected mice. The authors show that ONX treatment enhances early release of neutrophils and monocytes into the blood / spleen and enhances phagocytic function of these cells. Both chemokine and cytokine release, and importantly, infiltration of myeloid cells into heart tissue are reduced. Viral titers are less and cardiac function is improved, leading to the conclusion that systemic immunoproteasome (LMP7) inhibition by ONX0914 may represent an attractive means for treatment of severe virus-mediated inflammation of the heart.

These findings directly contradict a previous publication by the same authors, demonstrating that 'impairment of immunoproteasome function by LMP7 subunit deficiency results in severe enterovirus myocarditis' (Opitz et al, 2011). In this publication, it is shown that immunoproteasome deficiency is associated with severe acute heart muscle injury reflected by large foci of inflammatory lesions and severe myocardial damage. This injury is attributed to proteotoxic stress, resulting from impaired removal of poly-ubiquitinated protein aggregates in the absence of immunoproteasomes. In the mouse model used in this publication, proteasomes possessed six functional catalytic sites (LMP7 replaced by beta 5), while in the present manuscript, two of the catalytic sites of immunoproteasomes are blocked by ONX 0914.

Concern 1: This discrepancy between data in the present compared to the previous manuscript should be carefully explained.

Response: We are in perfect agreement with the reviewer that a careful explanation on these discrepant aspects of immunoproteasome function is mandatory and we apologize that this aspect was not adequately addressed in the original manuscript. In the revised manuscript, we highlight/report/demonstrate:

- The biological relevance of diverse hereditary susceptibility to CVB3 myocarditis in B6 and A/J mice as illustrated in the first paragraph of this letter.
- Disease course in ONX 0914-treated B6 mice (Fig 1) and detection of mildly increased inflammation being consistent with our report on B6-LMP7-deficient mice where myocardial virus-induced injury was exacerbated (Opitz et al, 2011).
- Diverse functional aspects of LMP7-deletion and ONX 0914-induced effects in B6 mice. This might be attributed to the fact that – as already stated by reviewer 2 – LMP7 deletion results in replacement with other standard proteasome catalytic subunits (Opitz et al, 2011; Paeschke et al, 2016) and ONX 0914 specifically blocks the LMP7 subunits (Huber et al, 2012; Muchamuel et al, 2009; Paeschke et al, 2016; Spur et al, 2016). Such aspects might contribute to controversial findings in B6 mice: e.g. similar type I IFN response in B6-LMP7 as compared to wild-type controls (Opitz et al, 2011) vs. suppression of type I IFN responses in ONX 0914-treated B6 mice upon virus inoculation (Fig 1F) which is in line with previous reports by other groups (Ichikawa et al, 2012; Muchamuel et al, 2009).

- In the supplemental material, we demonstrate accumulation of poly-ubiquitinated proteins preferentially within and around inflammatory foci in infected mouse hearts in A/J mice. Moreover, as expected, the overall abundance of such proteins was increased in mice with increased infiltration. Consistent with diminution of infiltration under ONX 0914 influence, ubiquitin-positive signals appeared to be reduced. This observation challenges a generalization of the previously proposed function of the immunoproteasome which is preservation of proteostasis under inflammatory conditions (Opitz et al, 2011; Seifert et al, 2010). The role of facilitated proteolysis by the immunoproteasome is evident in B6-LMP7 mice (Opitz et al, 2011; Seifert et al, 2010), but as demonstrated here other aspects predominate if specific inhibitors of the immunoproteasome are used or another genetic background is involved.

More detailed information is provided in the first section (overall achievements):

(A) Concerns on contradictory findings of CVB3-mediated acute myocarditis in B6-LMP7^{-/-} mice.

These various aspects are demonstrated and discussed accordingly in the revised manuscript.

Concern 2: In addition, any mechanistic insight into how described differences in innate immune cell presence and function, between ONX-treated and untreated infected mice, influence CVB3-caused pathology is lacking. If claiming an explanatory role for neutrophils and monocytes in the blood in early control of CVB3 infection in ONX-treated mice, additional experiments, for example with conditional KO mice, are needed to demonstrate the suggested role of these cells in amelioration of CVB3-induced disease.

Response: We have conducted several additional experiments that addressed the question whether and how ONX 0914 might influence direct virus-mediated pathology in A/J mice. Since ONX 0914 had no effect on the viral burden in heart and pancreas at the respective organ-specific peak of viral cytotoxicity and delayed virus dissemination as found 2 days after infection in heart, serum and spleen was attributed to diminished immunopathology in pancreatic tissue (Fig 3, results page 6, last paragraph), we concluded that ONX 0914-induced protection from virus-induced pathology in A/J mice did not involve a major antiviral effect.

Also, we have directly investigated the biological significance of neutrophils during CVB3 infection. NETosis, which has also been demonstrated to combat viruses, had no influence on CVB3 infection (Fig 5G). Depletion of neutrophils prior to infection in A/J mice neither influenced disease course nor did it influence viral burden in heart, pancreas and spleen (Fig 5H+EV5). We concluded that ONX 0914-induced neutrophilia most likely does not attribute to diminished immunopathology in A/J mice.

Regarding this aspect, please also refer to the first section on overall achievements of this letter:

(B) Concerns on the purported anti-viral function of ONX 0914.

Although inflammatory monocytes are essential early responders, their excessive or prolonged recruitment to the heart hinders resolution of inflammation and propagates disease progression (Hirasawa et al, 1996; Jaquenod De Giusti et al, 2015). ONX 0914 independently of virus inoculation mobilized inflammatory monocytotic cells from bone marrow sources and increased their abundance in spleen. Since ONX 0914-induced protection from virus-induced pathology in A/J mice did not involve a major antiviral effect, we investigated whether cytokine/chemokine expression, which is needed to induce migration and infiltration of inflammatory monocytotic cells from spleen into heart tissue, is affected by ONX 0914. Despite increased ONX 0914-induced infiltration of inflammatory monocytes, which majorly contribute to cytokine production into splenic tissue, the expression of respective pro-inflammatory cytokines/chemokines was suppressed under ONX 0914 influence (Fig 7A). Both global as well as heart-tissue specific suppression of pro-

inflammatory cytokine/chemokine production (Fig 7) and thereby mediated effects (Table 1, Fig 2) substantially contribute to suppression of detrimental adverse immune response activation and overall beneficial effects exerted by ONX 0914.

A detailed description of involved effector pathways is provided in the first section on overall achievements of this letter:

(D) Concerns on molecular insights into the protective influence exerted by ONX 0914 in CVB3-infected A/J mice.

More information on mechanistic insights of the respective consequences is provided under the section:

(C) Concerns on lack of demonstration that immune-mediated injury is reduced after ONX 0914 treatment in A/J mice.

Minor comment 1: B6 mice: please show weights of treated and untreated mouse groups over the full course of CVB3 challenge.

Response: The requested data is shown in Fig 1B.

Minor comment 2: Neutrophil / monocyte release from the BM is caused by chemokines. Please show whether serum levels of responsible chemokines are enhanced by ONX treatment.

Response: We have investigated various mediators (MIP-2, M-CSF, IL-1b, IL-4, GM-CSF, IFN-g, TNF-a, IL-6, MIP-1a, MCP-1, IL-23, IL-22, IL-33, IL-17, IFN-b), some of them being involved in the release of myeloid cells from bone marrow (both after single and 3 day ONX 0914 treatment). None of these mediators was increased by ONX 0914 in the absence of virus inoculation.

Minor comment 3: Ref to Fig 3A, B at p. 8 is misplaced.

Response: The concern has been addressed accordingly.

Reviewer 3:

Overall, experiments were executed very well and all the experiments were appropriate to support the claims by authors: highly relevant model and assays. The role of immunoproteasome in viral myocarditis is highly interesting and authors used a LMP7 inhibitor to test it. All the experiments were performed well with appropriate controls and the results obtained were very supportive of important role of immunoproteasome in the disease model. An additional control, in addition to captisol, that might further convince the role of LMP7 in viral myocarditis is the use of inactive ONX 0914 (an epimer with different stereochemistry at the epoxide ring). While the use of epimer may further convince the findings described in this manuscript, the work overall is well done and worth publication without revision.

Response: We kindly acknowledge the recommendation of reviewer 3 to publish our manuscript. We are grateful for the recommendation to include inactivated ONX 0914 as an additional control and will be happy to do so in future research.

References

Althof N, Harkins S, Kemball CC, Flynn CT, Alirezaei M, Whitton JL (2014) In vivo ablation of type I interferon receptor from cardiomyocytes delays coxsackieviral clearance and accelerates myocardial disease. *J Virol* 88: 5087-5099

Blasius AL, Beutler B (2010) Intracellular toll-like receptors. *Immunity* 32: 305-315

Cain BS, Meldrum DR, Dinarello CA, Meng X, Joo KS, Banerjee A, Harken AH (1999) Tumor necrosis factor-alpha and interleukin-1beta synergistically depress human myocardial function. *Critical care medicine* 27: 1309-1318

Cooper LT (2009) Medical Progress: Myocarditis. *New Engl J Med* 360: 1526-1538

Corsten MF, Schroen B, Heymans S (2012) Inflammation in viral myocarditis: friend or foe? *Trends Mol Med* 18: 426-437

Ebstein F, Voigt A, Lange N, Warnatsch A, Schroter F, Prozorovski T, Kuckelkorn U, Aktas O, Seifert U, Kloetzel PM et al (2013) Immunoproteasomes Are Important for Proteostasis in Immune Responses. *Cell* 152: 935-937

Esfandiarei M, McManus BM (2008) Molecular biology and pathogenesis of viral myocarditis. *AnnuRevPathol* 3: 127-155

Hensley SE, Zanker D, Dolan BP, David A, Hickman HD, Embry AC, Skon CN, Grebe KM, Griffin TA, Chen WS et al (2010) Unexpected Role for the Immunoproteasome Subunit LMP2 in Antiviral Humoral and Innate Immune Responses. *J Immunol* 184: 4115-4122

Hirasawa K, Tsutsui S, Takeda M, Mizutani M, Itagaki S, Doi K (1996) Depletion of Mac1-positive macrophages protects DBA/2 mice from encephalomyocarditis virus-induced myocarditis and diabetes. *J Gen Virol* 77 (Pt 4): 737-741

Huber EM, Basler M, Schwab R, Heinemeyer W, Kirk CJ, Groettrup M, Groll M (2012) Immuno- and Constitutive Proteasome Crystal Structures Reveal Differences in Substrate and Inhibitor Specificity. *Cell* 148: 727-738

Ichikawa HT, Conley T, Muchamuel T, Jiang J, Lee S, Owen T, Barnard J, Nevarez S, Goldman BI, Kirk CJ et al (2012) Beneficial effect of novel proteasome inhibitors in murine lupus via dual inhibition of type I interferon and autoantibody-secreting cells. *Arthritis Rheum* 64: 493-503

Jakel S, Kuckelkorn U, Szalay G, Plotz M, Textoris-Taube K, Opitz E, Klingel K, Stevanovic S, Kandolf R, Kotsch K et al (2009) Differential Interferon Responses Enhance Viral Epitope Generation by Myocardial Immunoproteasomes in Murine Enterovirus Myocarditis. *American Journal of Pathology* 175: 510-518

Jaquenod De Giusti C, Ure AE, Rivadeneyra L, Schattner M, Gomez RM (2015) Macrophages and galectin 3 play critical roles in CVB3-induced murine acute myocarditis and chronic fibrosis. *Journal of molecular and cellular cardiology* 85: 58-70

Kanda T, McManus JE, Nagai R, Imai S, Suzuki T, Yang D, McManus BM, Kobayashi I (1996) Modification of viral myocarditis in mice by interleukin-6. *Circ Res* 78: 848-856

Kindermann I, Kindermann M, Kandolf R, Klingel K, Bultmann B, Muller T, Lindinger A, Bohm M (2008) Predictors of outcome in patients with suspected myocarditis. *Circulation* 118: 639-648

Klingel K, Hohenadl C, Canu A, Albrecht M, Seemann M, Mall G, Kandolf R (1992) Ongoing Enterovirus-Induced Myocarditis Is Associated with Persistent Heart-Muscle Infection - Quantitative-Analysis of Virus-Replication, Tissue-Damage, and Inflammation. *Proceedings of the National Academy of Sciences of the United States of America* 89: 314-318

Kumar A, Thota V, Dee L, Olson J, Uretz E, Parrillo JE (1996) Tumor necrosis factor alpha and interleukin 1beta are responsible for in vitro myocardial cell depression induced by human septic shock serum. *J Exp Med* 183: 949-958

Muchamuel T, Basler M, Aujay MA, Suzuki E, Kalim KW, Lauer C, Sylvain C, Ring ER, Shields J, Jiang J et al (2009) A selective inhibitor of the immunoproteasome subunit LMP7 blocks cytokine production and attenuates progression of experimental arthritis. *Nat Med* 15: 781-787

Opitz E, Koch A, Klingel K, Schmidt F, Prokop S, Rahnefeld A, Sauter M, Heppner FL, Volker U, Kandolf R et al (2011) Impairment of immunoproteasome function by beta5i/LMP7 subunit deficiency results in severe enterovirus myocarditis. *PLoS Path 7*: 1-13

Paeschke A, Possehl A, Klingel K, Voss M, Voss K, Kesphohl M, Sauter M, Overkleeft HS, Althof N, Garlanda C et al (2016) The immunoproteasome controls the availability of the cardioprotective pattern recognition molecule Pentraxin3. *Eur J Immunol 46*: 619-633

Rahnefeld A, Klingel K, Schuermann A, Diny NL, Althof N, Lindner A, Bleienheuft P, Savvatis K, Respondek D, Opitz E et al (2014) Ubiquitin-Like Protein ISG15 (Interferon-Stimulated Gene of 15 kDa) in Host Defense Against Heart Failure in a Mouse Model of Virus-Induced Cardiomyopathy. *Circulation 130*: 1589-1600

Sagar S, Liu PP, Cooper LT, Jr. (2012) Myocarditis. *Lancet 379*: 738-747

Seifert U, Bialy LP, Ebstein F, Bech-Otschir D, Voigt A, Schroter F, Prozorovski T, Lange N, Steffen J, Rieger M et al (2010) Immunoproteasomes Preserve Protein Homeostasis upon Interferon-Induced Oxidative Stress. *Cell 142*: 613-624

Spur EM, Althof N, Respondek D, Klingel K, Heuser A, Overkleeft HS, Voigt A (2016) Inhibition of chymotryptic-like standard proteasome activity exacerbates doxorubicin-induced cytotoxicity in primary cardiomyocytes. *Toxicology 353-354*: 34-47

Szalay G, Meiners S, Voigt A, Lauber J, Spieth C, Speer N, Sauter M, Kuckelkorn U, Zell A, Klingel K et al (2006) Ongoing coxsackievirus myocarditis is associated with increased formation and activity of myocardial immunoproteasomes. *American Journal of Pathology 168*: 1542-1552

Tanaka T, Kanda T, McManus BM, Kanai H, Akiyama H, Sekiguchi K, Yokoyama T, Kurabayashi M (2001) Overexpression of interleukin-6 aggravates viral myocarditis: impaired increase in tumor necrosis factor-alpha. *Journal of molecular and cellular cardiology 33*: 1627-1635

2nd Editorial Decision

23 November 2017

Thank you for the submission of your revised manuscript to EMBO Molecular Medicine. We have now received the enclosed reports from the referees who were asked to re-assess it. As you will see the reviewers are now globally supportive and I am pleased to inform you that we will be able to accept your manuscript pending the following final amendments:

1) Please address Referee 1's comments in the text and provide a letter INCLUDING the reviewer's reports and your detailed responses to their comments (as Word file).

***** Reviewer's comments *****

Referee #1 (Remarks for Author):

Althof and colleagues did a great job to improve their manuscript. They added new experiments to support their hypothesis and changed the structure of their manuscript, which is now more linear and easier to understand. In general, there are still several weaknesses in the explanation of the protective mechanism conferred by ONX 0914 in A/J mice after CVB3 infection. I have some questions and comments on this revised manuscript.

1 - The authors observe that Granzyme A, Perforin, and IFN-gamma are reduced in the heart tissue at the RNA level in mice treated with ONX 0194 (it is not clear at which day the authors measured them). Since these three factors are related to the cytotoxic anti-viral immune response, how would

the author explain reduced viral titres at day 2 in ONX 0914-treated A/J mice?

Did the author determine the levels of cytotoxic CD8+ T cells and NK cells in the heart? It would be interesting to correlate the amount of Granzyme A, Perforin, and IFN-gamma together with cytotoxic cells at an early time point, such as at day 3, before the mice start dying.

2 - In Figure 3B no significant difference ($p=0.1421$) between vehicle-treated and ONX 0914-treated mice has been observed in terms of CVB3 genome in the heart at day 2. Why this discrepancy between the Plaque Assay in Figure 3A and the RNA levels measured in Figure 3B? The authors should also try to better explain why they observed significant differences of CVB3 titres at day 2, but not before or after that day. How many times did the authors repeat this experiment?

3 - The authors found higher numbers of lymphocytes in the spleen of ONX 0914-treated mice after viral challenge. Does ONX 0914 reduce apoptosis among lymphocytes or does it promote their proliferation? In Figure 4D and 4E, according to the figure legend, the first one represents an automatic complete blood count, while the other one is an analysis by flow cytometry. Why do the automatic blood count show significant differences among CD3 T cells (CD4 and CD8), while the FACS analysis does not show any significant differences? Is the description of the Y-axis maybe wrong?

4 - The analysis of Ly6C-high monocytes in Figure 6D should be better described in the figure legend. Did the author find an increase in Ly6C-high monocytes also at day 2 or day 3 post CVB3 infection? Did the author checked the amount of heart infiltrating monocytes at day 2 and day 8 p.i.?

5 - Figure 8B: there is a trend, but the differences are minimal (around 1:2 reduction). Are these differences big enough to explain the full protection of ONX 0914-treated A/J mice against the CVB3?

6 - The phosphorylation of MAP Kinases in ONX 0914-treated BMM after RNA incubation is slightly reduced. It is really difficult to see differences between ONX 0914 treatment and no treatment on the Western Blot. It is quite surprising that the densitometric analysis of the Western Blot in Figure 8D shows significant differences. This analysis with the densitometre has to be clearly explained.

7 - Minor comment: Point A of the explanation about the general improvement to the manuscript was not necessary and is a bit snotty, since the reviewers should already know what the authors describe in Point A. Otherwise the current reviewers would not have been selected as reviewers for this manuscript.

Referee #2 (Comments on Novelty/Model System for Author):

In response to my comments after initial review of this manuscript, the authors have performed a substantial number of new experiments to address my remarks. In this resubmission, they have explained their findings carefully, in context of previous publications. These changes raise my scores on all items.

Referee #2 (Remarks for Author):

In response to initial review of this manuscript, the authors have performed a substantial number of new experiments. In this resubmission, they have explained their findings carefully, in context of previous publications. Thereby, they have adequately addressed my prior points of criticism.

Referee #1 (Remarks for Author):

Althof and colleagues did a great job to improve their manuscript. They added new experiments to support their hypothesis and changed the structure of their manuscript which is now more linear and easier to understand. In general, there are still several weaknesses in the explanation of the protective mechanism conferred by ONX 0914 in A/J mice after CVB3 infection. I have some questions and comments on this revised manuscript.

Response: Once more, we kindly acknowledge the comprehensive review of our manuscript by the reviewer. Please find below a detailed point-to-point response regarding your questions and comments on the revised manuscript.

1 - The authors observe that Granzyme A, Perforin, and IFN-gamma are reduced in the heart tissue at the RNA level in mice treated with ONX 0194 (it is not clear at which day the authors measured them). Since these three factors are related to the cytotoxic anti-viral immune response, how would the author explain reduced viral titres at day 2 in ONX 0914-treated A/J mice?

Did the author determine the levels of cytotoxic CD8⁺ T cells and NK cells in the heart? It would be interesting to correlate the amount of Granzyme A, Perforin, and IFN-gamma together with cytotoxic cells at an early time point, such as at day 3, before the mice start dying.

Response: Granzyme A, perforin and IFN- γ were determined 8 days after infection in whole heart tissue homogenates. There was no discrimination of granzyme A, perforin and IFN- γ induction between CD8⁺ T cells and NK cells in the heart. From our point of view, our results on these parameters reflect reduced activation of immune cell-mediated cytotoxicity upon ONX 0914 treatment. Viral titers were not affected after 8 days in heart tissue. Moreover, the slight reduction of the virus load 2 days after infection in heart tissue is most likely not attributed to this reduced activation status found at day 8. We argued that a temporarily reduced viral burden in heart, spleen and serum at day 2 p.i. most likely reflects slightly altered viremia, which in turn might be the result of a diminished immune-mediated destruction of pancreatic tissue. We found altered signs of pancreas destruction such as reduced serum lipase activity in ONX 0914-treated mice (Fig EV1B/C). Analysis of infiltration in heart tissue at day 3 p.i. revealed very low abundance of immune cells. Therefore, we did not follow this point in time regarding inflammation in heart tissue and rather focused on the inflammation peak at day 8 p.i.

2 - In Figure 3B no significant difference ($p=0.1421$) between vehicle-treated and ONX 0914-treated mice has been observed in terms of CVB3 genome in the heart at day 2. Why this discrepancy between the Plaque Assay in Figure 3A and the RNA levels measured in Figure 3B? The authors should also try to better explain why they observed significant differences of CVB3 titres at day 2, but not before or after that day. How many times did the authors repeat this experiment?

Response: The statistical analysis of the parameter at day 2 p.i. in 3A determines a p-value of 0.003 and in 3B of 0.142. Nevertheless, there is a reduction of viral load at the level of viral titer: >1-log fold (3A) and viral genome: >50% (3B). In line with published data (Rahnefeld et al, 2014; PMID: 25165091), viral genome detection was quite variable during CVB3 Nancy infection as found 2 and 8 days after infection. This fact attributes to the respective p-value. Concerning our explanation on unaltered viral titers at all other states of infection, we would like to refer to comment 1 on pancreatic tissue destruction days 2 p.i. and all other data on unaltered T1IFN responses in infected A/J mice under ONX 0914 treatment. Sample size at day 2 p.i. \rightarrow 3A: n=9 vehicle; n=8 ONX; 3B: n=7 vehicle, n=8 ONX.

3 - The authors found higher numbers of lymphocytes in the spleen of ONX 0914-treated mice after viral challenge. Does ONX 0914 reduce apoptosis among lymphocytes or does it promote their proliferation? In Figure 4D and 4E, according to the figure legend, the first one represents an automatic complete blood count, while the other one is an analysis by flow cytometry. Why do the

automatic blood count show significant differences among CD3 T cells (CD4 and CD8), while the FACS analysis does not show any significant differences? Is the description of the Y-axis maybe wrong?

Response: Spleen lymphocyte count both at day 2 and 8 p.i. was determined by flow cytometry. Figure 4C represents the results in blood and these data were obtained from automated blood counts. The figure legend was revised regarding this aspect. Data in 4D represent changes 2 days after infection and data in 4E 8 days after infection.

4 - The analysis of Ly6C-high monocytes in Figure 6D should be better described in the figure legend. Did the author find an increase in Ly6C-high monocytes also at day 2 or day 3 post CVB3 infection? Did the author checked the amount of heart infiltrating monocytes at day 2 and day 8 p.i.?

Response: We included a link referring to the respective strategy: "A detailed description of the gating strategy for identification of the different immune cell type is provided in Fig EV2." Figure 6C displays myeloid cell abundance in spleen 3 days after infection. For heart tissue, we stained CD45⁺/CD11b^{high} for myeloid cells 8 days after infection (Fig 2E). Also, macrophage and monocyte populations were defined showing the same results of reduced infiltration of the respective population. 3 days after infection, we detected a very low abundance of monocytes/macrophages and no alteration between both groups. In other words, at this state we found no significant infiltration. Since these data provided no additive information and were in line with expected results, we decided not to include them in the manuscript.

5 - Figure 8B: there is a trend, but the differences are minimal (around 1:2 reduction). Are these differences big enough to explain the full protection of ONX 0914-treated A/J mice against the CVB3?

Response: A reduction of 50% of various pro-inflammatory cytokines / chemokines with known systemic impact on various tissues and hemodynamic compromise appears to be a plausible explanation for the observed effects induced by ONX 0914. The reviewer refers to Fig 8. This figure displays *in vitro* data from BMM stimulated with a TLR7 ligand. A conclusion on the definite *in vivo* impact of these specific *in vitro* results is not possible.

6 - The phosphorylation of MAP Kinases in ONX 0194-treated BMM after RNA incubation is slightly reduced. It is really difficult to see differences between ONX 0914 treatment and no treatment on the Western Blot. It is quite surprising that the densitometric analysis of the Western Blot in Figure 8D shows significant differences. This analysis with the densitometre has to be clearly explained.

Response: We have been using the Odyssey® CLx Imaging System (Li-Cor Biosciences) in conjunction with near-infrared fluorescence secondary antibodies [IRDye800CW labelled goat anti-mouse (p-p38)/anti-rabbit (all remaining) secondary antibodies] to ensure sensitive, quantitative, and complete Western blot assessment. This technology delivers consistent, reproducible digital images and is particularly useful for quantitative measurements as demonstrated here. Signal intensity at 15 min was determined for DMSO- and ONX 0914-treated cells by Li-Cor software and signals of phosphorylated kinases (e.g. p-p38 and p-p42) were normalized to the respective non-phosphorylated protein.

For 3 independent experiments, thereby obtained data of ONX 0914-treated cells were normalized to respective DMSO controls (15 min R848 stimulus) and plotted bar graphs demonstrate mean of the respective phosphorylated MAPkinases. Example for p-p42: experiment 1 – p-p42 signal in ONX 0914-treated cells revealed 81% of the signal intensity found in DMSO-treated controls at 15 min; experiment 2 – ONX 0914-treated cells revealed 69% of the signal intensity of DMSO-treated controls at 15 min, experiment 3 – ONX 0914-treated cells revealed 73% of the signal intensity of DMSO-treated controls at 15 min. Average of 3 independent experiments of the p-p42 signal in ONX 0914-treated cells: 74%).

7 - Minor comment: Point A of the explanation about the general improvement to the manuscript was not necessary and is a bit snotty, since the reviewers should already know what the authors

describe in Point A. Otherwise the current reviewers would not have been selected as reviewers for this manuscript.

Response: We regret that the general introductory part on the different genotypes needed to discuss our previous findings in B6-LMP7 mice and the different biological function of the immunoproteasome in A/J mice was unnecessarily long.

Corresponding Author Name: Prof. Dr. med. Antje Beling

Journal Submitted to: EMBO Mol Med

Manuscript Number: EMM-2017-08089